# Revolutionizing Reinforcement Learning Framework for Diffusion Large Language Models

**Yinjie Wang**[1,2*], **Ling Yang**[1*†], **Bowen Li**[3], **Ye Tian**[3], **Ke Shen**, **Mengdi Wang**[1]

[1]Princeton University, [2]University of Chicago, [3]Peking University

Project: https://github.com/Gen-Verse/dLLM-RL

## Abstract

The extension of diffusion models to language tasks has shown promising results, but their post-training methods remain largely unexplored. We highlight the importance of aligning a diffusion language model's preference-inference trajectory with its post-training objective. To this end, we propose **TraceRL**, a trajectory-aware reinforcement learning framework for DLMs that incorporates information from inference trajectories into post-training and is applicable to both full-attention and block-attention diffusion models. We also introduce a diffusion-based value model that enhances training stability and naturally accommodates process rewards. We demonstrate TraceRL's superiority in enhancing a model's reasoning ability on complex math and coding tasks, as well as its applicability in scaling block diffusion models to larger block sizes. Employing TraceRL, we derive a series of state-of-the-art diffusion language models, namely **TraDo**. Although smaller than Qwen2.5-7B-Instruct, TraDo-4B-Instruct consistently outperforms it on complex math reasoning tasks. TraDo-8B-Instruct achieves 4.5% higher accuracy on MATH500 than Qwen2.5-7B-Instruct and 6.6% higher accuracy on LiveCodeBench-V2 than Llama3.1-8B-Instruct. Through curriculum learning, we also develop the first 8B-scale long-CoT diffusion language model.

## 1 Introduction

To apply diffusion models (Ho et al., 2020; Song & Ermon, 2019) on language tasks, approaches such as projecting discrete tokens into a continuous space (Graves et al., 2023; Dieleman et al., 2022; Gulrajani & Hashimoto, 2023), and employing state transition matrices to derive the diffusion process (Austin et al., 2021; Sahoo et al., 2024; Shi et al., 2024; Ou et al., 2024) have been explored. Masked diffusion models (MDMs) have emerged as a promising and scalable architecture for diffusion language models (DLMs) recently (Nie et al., 2025; Ye et al., 2025). DLMs enable substantial inference acceleration by parallel generation (Labs et al., 2025; Google DeepMind, 2025; Song et al., 2025), and improve consistency through bidirectional attention (Zhang et al., 2023).

Diffusion language models with full attention have been explored in coding tasks (Labs et al., 2025; Google DeepMind, 2025; Gong et al., 2025; Xie et al., 2025) and in fixed-format reasoning tasks (Ye et al., 2024; 2025). Beyond the full attention mechanism, block-attention diffusion models (Arriola et al., 2025) scaled up to large language models that excel at complex reasoning tasks (Cheng et al., 2025). However, a unified and effective reinforcement learning (RL) framework applicable to both full-attention and block-attention DLMs remains largely unexplored.

Existing post-training frameworks focus on full-attention DLMs by optimizing the entire sampled sequence with randomly added masks. This approach leads to a mismatch with the optimal inference process (Ye et al., 2025; Nie et al., 2025; Gong et al., 2025; Yang et al., 2025b), since the sequential and logical nature of language is not purely random (Arriola et al., 2025; Gong et al., 2025). We first demonstrate the importance of aligning the inference trajectory with the training objective

---

*Equal Contribution. Contact: yangling0818@163.com

†Corresponding Author

through empirical studies, and then propose a trajectory-aware reinforcement learning method, TraceRL, which is applicable on both full-attention and block-attention DLMs. TraceRL demonstrates stronger optimization performance than baseline methods across diverse DLMs and tasks. Notably, optimizing the math reasoning ability of a 4B model yields a 5.4% improvement in accuracy on MATH500, outperforming Qwen2.5-7B-Instruct. We also introduce a diffusion-based value model that reduces variance and enhances training stability. Furthermore, we show that this value model naturally accommodates the process rewards, providing stronger reward supervision.

Trained solely with TraceRL, we obtain 4B and 8B state-of-the-art diffusion language instruction models on reasoning tasks, both surpass strong autoregressive models on math reasoning benchmarks. Furthermore, we develop the first 8B-scale long-CoT diffusion language model, TraDo-8B-Thinking, by combining TraceRL with long-CoT SFT.

We summarize our main contributions as follows:

- We demonstrate the importance of aligning the training objective with the inference trajectory through empirical experiments. Building on this insight, we propose TraceRL, a trajectory-aware RL method applicable to both full-attention and block-attention DLMs.
- We introduce a diffusion-based value model that reduces variance and enhances training stability via a token-wise baseline and GAE-based targets optimized using a clipped regression objective. This value model also naturally accommodates process rewards, providing more nuanced reward supervision.
- The superiority of TraceRL is demonstrated across multiple architectures and tasks. Optimizing a 4B model on math tasks yields 5.4% higher accuracy on MATH500, surpassing Qwen2.5-7B-Instruct, while optimizing an 8B model achieves 7.4% higher accuracy on LiveCodeBench-V2. TraceRL also scales block diffusion models to larger block sizes.
- Using TraceRL alone, we obtain state-of-the-art diffusion language models, the TraDo series. TraDo-8B-Instruct achieves 6.1% higher accuracy than Qwen2.5-7B-Instruct on MATH500 and 5.9% higher accuracy than Llama-3.1-8B-Instruct on LiveCodeBench-V2. Furthermore, through curriculum learning, we develop the first 8B-scale long-CoT DLM.

## 2 RELATED WORK

### 2.1 MASKED DIFFUSION LANGUAGE MODELS WITH FULL ATTENTION

Masked diffusion language models (Nie et al., 2025; Yang et al., 2025b) randomly replace a subset of tokens in the raw sample $x_0$ with mask tokens [MASK], resulting in $x_t$, where $t \in [0, 1]$. This process can be formulated as Equation 1. Its reverse process can be accelerated through iterative generation, where multiple masked tokens are approximately recovered in one step when transitioning from noise level $t$ to a lower level $s < t$.

$$q(x_t \mid x_0) = \prod_{i=1}^{n} \text{Cat}\Big(x_t^i; (1-t)\delta_{x_0^i} + t\delta_{[\text{MASK}]}\Big). \tag{1}$$

It has been demonstrated that the training objective of DLMs can be derived from data likelihood (Shi et al., 2024; Sahoo et al., 2024; Zheng et al., 2024; Ou et al., 2024). Specifically, the optimization objective is the evidence lower bound of $\log p_\theta(x)$:

$$\mathcal{J}_{full}(x_0, Q, \theta) = \int_0^1 \frac{1}{t|x_0|} \mathbb{E}_{q(x_t|x_0)} \left[ \sum_{i:x_t^i=[\text{MASK}]} \log p_\theta(x_0^i \mid x_t, Q) \right] dt, \tag{2}$$

where $Q$ denotes the prompt part, and $|x_0|$ denotes the number of tokens in $x_0$.

### 2.2 ADAPTED MODELS AND BLOCK DIFFUSION MODELS

Adapting from an autoregressive model has been shown to be a promising approach for preserving the AR model's capabilities (Ye et al., 2025). This approach is efficient and has led to powerful DLMs (Ye et al., 2025; Gong et al., 2025; Xie et al., 2025; Cheng et al., 2025).

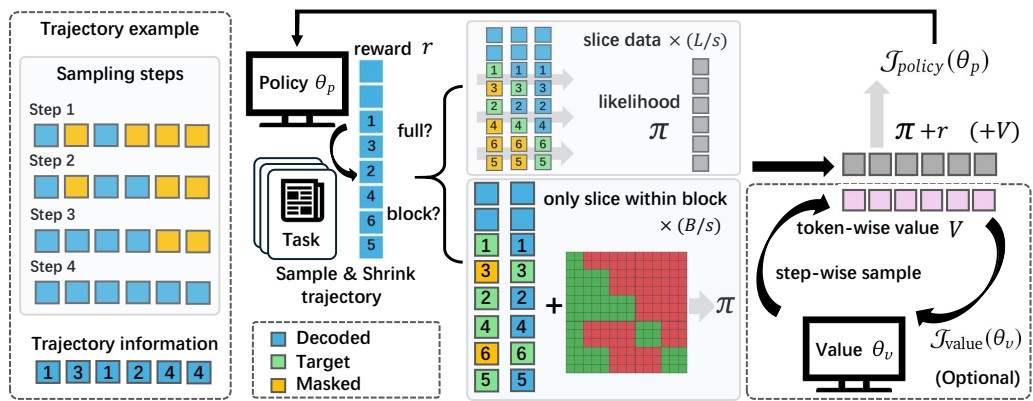

Figure 1: Trajectory illustration (left) and TraceRL overview (right, see details in Section 4). Right panel is an example of TraceRL with shrinkage parameter $s = 2$, sequence length $L = 6$, and block size $B = 3$ (when using block diffusion). We aggregate $s$ consecutive steps to perform trajectory-aware reinforcement learning. Integers inside the squares indicate trajectory information.

Block diffusion models (Arriola et al., 2025; Cheng et al., 2025) employ a block-diffusion attention mechanism that combines the training efficiency of autoregressive models with the sampling efficiency of diffusion models. Their block attention naturally supports KV-cache, mitigating the performance degradation often observed with KV-cache in full-attention DLMs (Hu et al., 2025b; Liu et al., 2025; Ma et al., 2025; Arriola et al., 2025; Yu et al., 2025b; Wu et al., 2025).

### 2.3 RL FOR DIFFUSION LANGUAGE MODEL

Reinforcement learning methods for full-attention models have been explored in prior work (Yang et al., 2025b; Gong et al., 2025; Zhao et al., 2025). In MMaDA, Yang et al. (2025b) introduce random masking for each rollout sample and optimize a PPO objective for training. Huang et al. (2025) treats each intermediate step in the reverse diffusion process as a latent thinking action and optimizes the entire reasoning trajectory. Gong et al. (2025) propose augmenting each rollout $x_0$ with both a single noise sample $x_t$ and its complementary counterpart $\widehat{x}_t$, thereby doubling the effective training data. They provide a theoretical analysis showing that this strategy reduces variance, yielding faster and more stable optimization than random-masking RL.

## 3 MISMATCH BETWEEN POST-TRAINING OBJECTIVE AND INFERENCE TRAJECTORY

Pretraining under the fully random masking objective (Equation 2) enables parallel decoding. However, natural language inherently depends on context. The widely adopted decoding strategy and KV-cache techniques create a mismatch between the post-training objective and the model's inference behavior. We present a simple experiment to illustrate this mismatch in this section.

### 3.1 INFERENCE TRAJECTORY

Strategies have been proposed to improve sampling quality. Given a masked sequence $x_t$, we select [MASK] tokens to unmask based on confidence at each position $i$, defined as the maximum predicted probability $\max_{x_s^i} p_\theta(x_s^i \mid x_t, Q)$. A common strategy is to unmask either a fixed number of tokens with the highest confidence (Chang et al., 2022; Kim et al., 2025), referred to as **static sampling** or all positions whose confidence exceeds a threshold $\mathcal{T}$ (Yu et al., 2025b; Wu et al., 2025), referred to as **dynamic sampling**, with the latter often faster. By iteratively applying this procedure starting from a fully masked sequence, we eventually recover a fully unmasked sequence.

### 3.2 SEMI-AUTOREGRESSIVE FINE-TUNING

Block diffusion models naturally employ semi-autoregressive fine-tuning with $\mathcal{J}_{\text{semi}}(x, Q, \theta)$ (Equation 7), which trains the model to generate later tokens conditioned on earlier context, while preserv-

Table 1: We explore how effectively different methods tune the model to learn CoT reasoning and thereby improve reasoning accuracy under non-CoT prompts. 2000 datapoints were generated using Qwen2.5 models and filtered for quality. $l$ denotes the length of each step in the complete trace. "$\times m$" indicates that we apply $m$ independent random maskings to augment the dataset for a fair comparison. "Token forward" denotes the number of tokens processed by the model, representing computational load/time. "Token trained" refers to the number of tokens directly contributing to the optimization objective. We report accuracy on MATH500. The block-attention model used here is SDAR-4B-Chat (block size of 4), and the full-attention model is Dream-7B-Instruct.

| Token Utilization Efficiency | Full Attention | | | | | Block Attention | | |
|---|---|---|---|---|---|---|---|---|
| | trace | semi-ar ($B = 16$) | | fully random | | trace | semi-ar ($B = 4$) | |
| | $l = 16$ | $\times 2$ | $\times 1$ | $\times 35$ | $\times 1$ | $l = 2$ | $\times 2$ | $\times 1$ |
| accuracy (%) | 54.4 | 53.4 | 52.6 | 45.1 | 39.6 | 71.3 | 70.4 | 70.0 |
| # token forward (M) | 39.2 | 78.4 | 39.2 | 39.2 | 1.1 | 4.5 | 4.5 | 2.2 |
| # token trained (M) | 1.1 | 1.1 | 0.6 | 20.0 | 0.6 | 1.1 | 1.1 | 0.6 |

ing the sampling efficiency of DLMs. For full-attention DLMs, applying the semi-autoregressive objective inevitably increases forward passes by about $\lceil L/B \rceil$ (see Figure 6). Nevertheless, the optimization performance of $\mathcal{J}_{\text{semi}}(x, Q, \theta)$ is substantially better than that of $\mathcal{J}_{\text{full}}(x, Q, \theta)$ under $\lceil L/B \rceil$ independent repetitions with the same computational load (see Table 1). This demonstrates the importance of aligning the post-training objective with the general left-to-right inference pattern.

## 3.3 ALIGNING POST-TRAINING WITH PREFERRED INFERENCE TRACES

To further investigate, we collect the pre-trained model (DLM)'s own preferred inference traces and use them for fine-tuning. Specifically, we apply static sampling (see definition in Section 3.1) to obtain the optimal trace for each data point, which is then used in our fine-tuning (see Appendix D.1 for details). Table 1 shows that using the model's own preferred inference traces achieves superior performance compared to the baselines, even when the computational load is equal to or lower than that of the baselines, for both block-attention and full-attention models. A major limitation of this fine-tuning approach is that collecting inference traces is computationally and practically costly. In contrast, reinforcement learning naturally produces these traces during rollouts, making it a practical and effective strategy for post-training. Accordingly, we propose TraceRL in the following section.

## 4 RL TRAINING WITH TRAJECTORY

Current RL methods for DLMs focus on full-attention models, reward or penalize rollout responses based on the overall sequence generated (Zhao et al., 2025; Yang et al., 2025b; Gong et al., 2025) using the random masking objective $\mathcal{J}_{\text{full}}$ (Equation 2). We propose **TraceRL** (Figure 1, detailed pipeline in Appendix D.7), which instead focuses on the intermediate traces generated by the DLM and can be applied across different architectures. We also introduce a diffusion-based value model, which improves training stability and naturally accommodates process rewards.

For each generated response $\tau_i$ given the task $Q$, we present it in the trajectory form $\tau_i \overset{\triangle}{=} \tau_i(1) \cup \cdots \cup \tau_i(|\tau_i|)$, where $|\tau_i|$ is the number of decoding steps, and $\tau_i(t)$ is the set of tokens decoded during the $t$-th step. TraceRL rewards or penalizes the sampling trajectory under policy $\pi_\theta$, based on the verifiable reward $r_i$ assigned to $\tau_i$. Process rewards can be incorporated via value model.

## 4.1 ACCELERATED TRAINING WITH SHRINKAGE PARAMETER

For full-attention models, decomposing the sequences based on each sampling step leads to a large number of forward passes during training. To address this, we introduce a shrinkage parameter $s$, which aggregates every $s$ consecutive steps to improve training efficiency. Formally, we compress the trajectory $\tau_i$ into $\tau_i^s \overset{\triangle}{=} \tau_i^s(1) \cup \cdots \cup \tau_i^s(|\tau_i^s|)$, where $\tau_i^s(k) \overset{\triangle}{=} \cup_{j=s(k-1)+1}^{\min(sk,|\tau_i|)} \tau_i(j)$ and $|\tau_i^s| =$

$\lceil |\tau_i|/s \rceil$, then optimize the following objective:

$$\mathcal{J}_{policy}(\theta_p) = \mathbb{E}_{\substack{Q \sim D_{\text{task}} \\ \{\tau_i\}_{i=1}^{G} \sim \pi_{\text{old}}(\cdot|Q)}} \Big[ \sum_{i=1}^{G} \sum_{t=1}^{|\tau_i^s|} \sum_{o_k \in \tau_{i,t}^s} C_\epsilon \big( \frac{\pi_{\theta_p}(o_k \mid \tau_i^s(1:(t-1)))}{\pi_{\text{old}}(o_k \mid \tau_i^s(1:(t-1)))}, A_i \big) / |\tau_i^s(t)| \Big] - \beta \mathbb{KL},$$

(3)

where $C_\epsilon(r, A) \triangleq \min(rA, \text{clip}(r, 1 - \epsilon, 1 + \epsilon)A)$, $\tau_i^s(1:t) \triangleq \cup_{j=1}^{t} \tau_i^s(j)$, $\pi_{\text{old}}$ is the old policy, $A_i$ is the standardized reward, and $\mathbb{KL} = \mathbb{KL}[\pi_\theta || \pi_{\text{old}}]$.

By introducing the shrinkage parameter $s$, we reduce training complexity by a factor of $s$. This allows $\mathcal{J}_{policy}(\theta_p)$ to be used for efficient RLVR (reinforcement learning with verifiable rewards).

## 4.2 VALUE MODEL WITH DIFFUSION MODELING

Rather than assigning a single sequence-level advantage to all tokens in a completion, a value function enables token-wise advantages conditioned on the prefix (e.g., GAE-based estimates), providing a variance-reducing baseline that typically stabilizes policy optimization (Schulman et al., 2017; Hu et al., 2025a). We adopt a diffusion-based value model to estimate step-wise values. For each rolled-out trajectory $\tau$ (possibly shrunken by a shrinkage factor $s$), we retain the notation $\tau$ for brevity. We evaluate a frozen value network $V_{\text{old}}$ to obtain token-wise values along the trajectory: $V_j^{\text{old}} \triangleq \big( V_{\text{old}}(\tau) \big)_j$, $j \in \tau$. Concretely, for each trace step $t$, we condition on the prefix $\tau(1:t-1)$ and predict the values for all tokens generated at that step, i.e., $\{ V_j^{\text{old}} : j \in \tau(t) \}$. We then define the step-wise value at step $t$ as $V_t^{\star,\text{old}} = \sum_{j \in \tau(t)} V_j^{\text{old}}/|\tau(t)|$. $V_j^{\text{old}}$ is treated as a *stop-gradient* baseline when constructing returns and advantages; the learnable value network $V_{\theta_v}$ is updated only via its own regression objective.

We now derive the training objective. Given token-wise rewards $r_j$, we first form the step-wise reward $r_t^\star = \sum_{j \in \tau(t)} r_j/|\tau(t)|$. We compute step-wise returns recursively as $R_t^\star = r_t^\star + \gamma R_{t+1}^\star$ for $t \leq |\tau|$ with terminal condition $R_{|\tau|+1}^\star = 0$. Using the frozen baseline, we define $V_t^{\star,\text{old}} = \sum_{j \in \tau(t)} V_j^{\text{old}}/|\tau(t)|$, the TD residual is $\delta_t^\star = r_t^\star - V_t^{\star,\text{old}} + \gamma V_{t+1}^{\star,\text{old}}$, and the step-wise GAE is $A_t^\star = \delta_t^\star + \gamma \lambda A_{t+1}^\star$. We then map step-level quantities back to tokens as $R_j = r_j + \gamma R_{t_j+1}^\star$ and $A_j = (r_j - V_j^{\text{old}}) + \gamma V_{t_j+1}^{\star,\text{old}} + \gamma \lambda A_{t_j+1}^\star$, where $t_j$ is such that $j \in \tau(t_j)$. The value network is trained using a clipped regression loss:

$$\mathcal{J}_{\text{value}}(\theta_v) = \tfrac{1}{2} \mathbb{E}_\tau \Big[ \tfrac{1}{|\tau|} \sum_{j \in \tau} \max \big( (V_{\theta_v}(\tau)_j - R_j)^2, (V_j^{\text{clip}} - R_j)^2 \big) \Big],$$

(4)

where $V_j^{\text{clip}} = V_j^{\text{old}} + \text{clip}\big( V_{\theta_v}(\tau)_j - V_j^{\text{old}}, -\epsilon, \epsilon \big)$. The token advantages $A_j$ are used in $\mathcal{J}_{\text{policy}}(\theta_p)$ to update the policy. The explicit forms of $A_j$ and the theoretical analysis are provided in Appendix A.

## 4.3 SLICED TRAINING IN BLOCK DIFFUSION

Block diffusion uses block attention for efficient supervised fine-tuning, and we extend this to reinforcement learning. For each processed trace $\tau$ generated by block-wise inference, we write $\tau = (b_1, \ldots, b_{\lceil |\tau|/B \rceil})$, where each block is $b_k = (\tau_{k,1}, \ldots, \tau_{k,|b_k|})$ with $|b_k|$ steps, and $B$ is block size. The training objective can then be sliced from $\sum_{i=1}^{|\tau|} f(\tau(i))$ into a $B'$ training slice, $\Big( \sum_{k=1}^{\lceil |\tau|/B \rceil} f(\tau_{k,l}) \mathbf{1}_{\{l \leq |b_k|\}} \Big)_{l=1}^{B'}$, where $f$ is a task-specific function and $B' = \max_k \{ |b_k| \} \leq \lceil B/s \rceil$. Each slice requires one forward pass with block attention (Figure 1). This formulation maximizes the utility of the block-attention mechanism and enables highly parallel and efficient training, for both policy and value models, significantly more efficient than full-attention training.

## 5 EXPERIMENTS

In this section, we demonstrate the superiority of TraceRL across diverse tasks and models, as well as the advantages of incorporating a value model, including enhanced training stability and the natural

Table 2: The main benchmark results across different math and coding tasks. "Static" denotes static sampling, and "Dynamic" denotes dynamic sampling. The long-CoT model TraDo-8B-Instruct is evaluated using dynamic sampling with threshold 0.9.

| Model | MATH500 | | AIME2024 | | GSM8K | | LiveCodeBench-v2 | | LiveBench | |
|---|---|---|---|---|---|---|---|---|---|---|
| **Autoregressive Models** | | | | | | | | | | |
| Llama3.1-8B-Instruct | 51.9 | | 6.7 | | 84.5 | | 20.0 | | 19.7 | |
| Qwen2.5-7B-Instruct | 74.0 | | 8.2 | | 89.9 | | 26.9 | | 31.1 | |
| **Diffusion Language Models** | | | | | | | | | | |
| | Static | Dynamic | Static | Dynamic | Static | Dynamic | Static | Dynamic | Static | Dynamic |
| LLaDA-8B-Instruct | 37.3 | 38.3 | 0.5 | 1.7 | 82.5 | 82.5 | 5.9 | 5.5 | 4.9 | 6.0 |
| Dream-7B-Instruct | 38.7 | 32.3 | / | / | 72.7 | 57.8 | 10.7 | 4.7 | 10.7 | 4.9 |
| SDAR-4B-Chat | 70.2 | 67.4 | 5.0 | 8.2 | 90.2 | 88.9 | 15.6 | 11.2 | 14.0 | 7.6 |
| **TraDo-4B-Instruct** | **75.6** $_{+5.4}$ | **71.8** $_{+4.4}$ | **8.3** $_{+3.3}$ | **10.3** $_{+2.1}$ | **91.2** $_{+1.0}$ | **90.3** $_{+1.2}$ | **18.7** $_{+3.1}$ | **15.1** $_{+3.9}$ | **12.9** | **10.4** $_{+2.8}$ |
| SDAR-8B-Chat | 74.3 | 70.7 | 11.8 | 8.3 | 91.1 | 90.4 | 18.5 | 15.3 | 11.5 | 11.2 |
| **TraDo-8B-Instruct** | **78.5** $_{+4.2}$ | **75.5** $_{+4.8}$ | **13.3** $_{+1.5}$ | **11.0** $_{+2.7}$ | **92.3** $_{+1.2}$ | **91.2** $_{+0.8}$ | **25.9** $_{+7.4}$ | **22.4** $_{+7.1}$ | **22.7** $_{+11.2}$ | **20.6** $_{+9.4}$ |
| **TraDo-8B-Thinking** | **87.4** $_{+13.1}$ | | **35.5** $_{+23.7}$ | | **94.2** $_{+3.1}$ | | **34.6** $_{+16.1}$ | | **36.0** $_{+23.8}$ | |

ability to accommodate process rewards for trajectory-wise supervision. We present comprehensive evaluation results for our state-of-the-art TraDo models and highlight interesting applications such as block size enlargement. More ablation studies are included in Appendix C.

## 5.1 EXPERIMENTAL SETUPS

### 5.1.1 DATA

We use different data sources for reinforcement learning. For the math tasks, we choose the MATH training dataset (Hendrycks et al., 2021) and retain only level 3–5 tasks (Hu et al., 2025a), resulting in 8K hard tasks. For the coding RL setting, we use 6K verified problems from PrimeIntellect (Jaghouar et al., 2024), validated by DeepCoder (Luo et al., 2025), as a coding dataset in stdio format. In addition, we randomly sample 5K problems in the functional-test format from the Open-Coder training set (Huang et al., 2024).

Our evaluation focuses on reasoning tasks in mathematics and coding. For mathematics, we use GSM8K (Cobbe et al., 2021), MATH500 (Hendrycks et al., 2021), and AIME2024 (Mathematical Association of America, 2024). For coding, we use LiveCodeBench-V2 (Jain et al., 2024) and LiveBench (White et al., 2024).

### 5.1.2 MODELS AND OPTIMIZATION

Our experiments include both full-attention and block-attention models. For full attention, we use Dream-7B-Instruct (Ye et al., 2025) as the base model, and for block attention, we use the SDAR (Cheng et al., 2025) series of models, trained with a block size of 4.

We now describe our reinforcement learning settings. For the block-diffusion model, at each sampling step we sample 128 tasks and 32 responses per task, using a dynamic sampling strategy ($\mathcal{T} = 0.9$) and temperature 1.0. For the full-attention model, during each RL sampling step we sample 56 tasks and generate 8 responses per task from the policy $\pi_\theta$ with KV-Cache, with a temperature of 1.0. During training, we set the learning rate of the policy to $1 \times 10^{-6}$, with $\epsilon = 0.2$ and $\beta = 0.01$. When using the value model, we set the RL learning rate to $5 \times 10^{-6}$, and use $\gamma = \lambda = 1.0$ as default values. Our 4B and 8B instruction models, TraDo, based on SDAR-4B-Chat and SDAR-8B-Chat, are first trained on math tasks to convergence, then on coding tasks to convergence, and finally retrained on math tasks. Additional optimization details are provided in the Appendix D.4 and D.5.

### 5.1.3 EVALUATION

For the block-attention model, we use temperature $t = 1.0$ with dynamic sampling and greedy decoding (top-$k = 1$ for static sampling), with a default block size of 4. For the full-attention model, we use temperature $t = 0.1$ and block size 32 for static sampling. For the LLaDA model, we adopt a block size of 32 for dynamic sampling, whereas for the Dream model, we use a block size of 4 for dynamic sampling. Additional evaluation details and prompt templates are provided in Appendix D.3 and D.2.

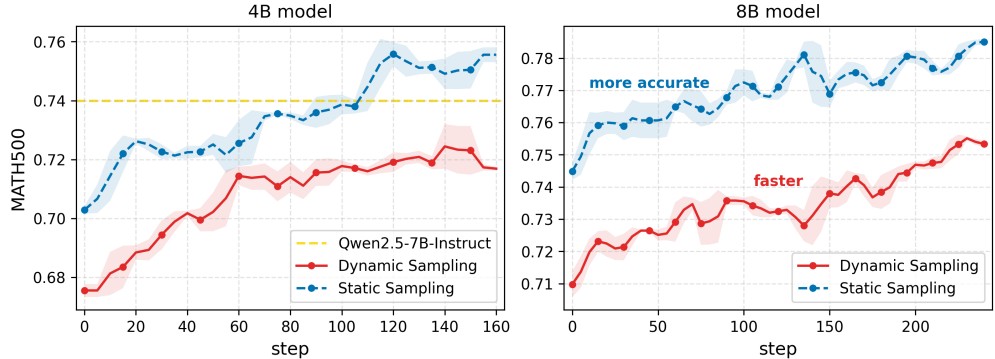

Figure 2: The TraceRL training curves for the 4B and 8B models on the math task. The red curve denotes the dynamic sampling accuracy, which yields faster sampling speed, while the blue curve denotes the static sampling accuracy, which yields higher accuracy. The 4B model is trained with a value model, whereas the 8B model is trained directly using $\mathcal{J}_{\text{policy}}$.

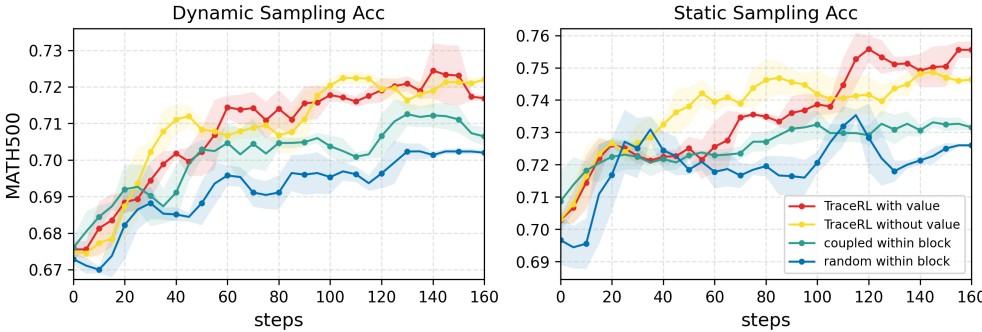

Figure 3: RL method ablations on block diffusion models for math RL tasks. The red and yellow curves represent TraceRL with and without a value model, respectively. The blue curve corresponds to training with a random masking objective in each block, similar to the semi-autoregressive approach. The green curve represents training with an additional complementary mask within block.

## 5.2 INSTRUCTION MODELS TRAINED WITH TRACERL

We obtain TraDo-4B-Instruct and TraDo-8B-Instruct by applying TraceRL to math and coding tasks, starting from the SDAR base model. We evaluate these models on math and coding tasks, as well as five reasoning datasets, and compare them against both strong diffusion language models (Cheng et al., 2025; Ye et al., 2025; Nie et al., 2025) and autoregressive models (Yang et al., 2024a; Dubey et al., 2024) (see Table 2). Our instruction models achieve state-of-the-art performance on reasoning tasks among open-source diffusion language models under both dynamic and static sampling, highlighting the effectiveness of TraceRL. Notably, the 4B model yields a 5.4% accuracy gain on MATH500, surpassing Qwen2.5-7B-Instruct across all math tasks. 8B model achieves a 7.4% gain on LiveCodeBench-V2, scoring 5.9% higher than Llama-3.1-8B-Instruct on the same benchmark.

## 5.3 TRAINING DYNAMICS OF TRACERL

We present the training dynamics on math tasks for our 4B and 8B instruction models (see Figure 2). Although dynamic sampling is adopted during RL training, both dynamic and static accuracy improve steadily, suggesting further potential for scaling. This RL training substantially enhances the models' math reasoning ability: on MATH500, TraDo-4B-Instruct improves by 5.4% (static) and 4.2% (dynamic), surpassing Qwen2.5-7B-Instruct after optimization, while TraDo-8B-Instruct improves by 4.2% (static) and 4.8% (dynamic) (see Figure 2).

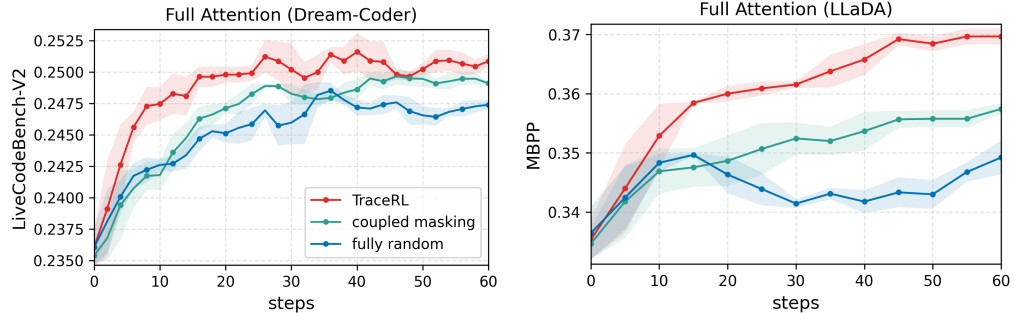

Figure 4: RL training ablations for the full-attention model on coding tasks. (a). Dream-7B-Coder-Instruct. (b) LLaDA-8B-Instruct.

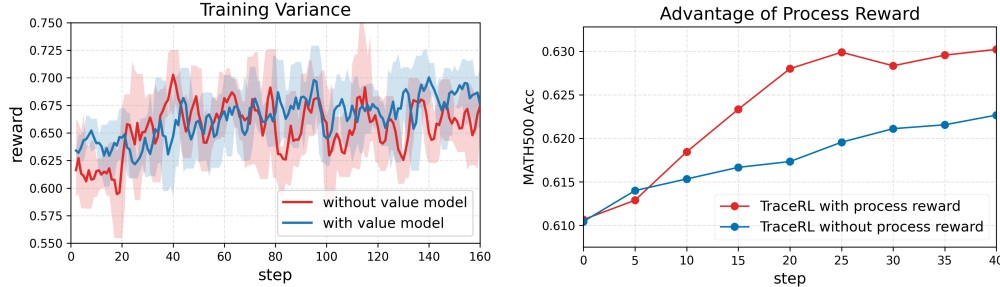

Figure 5: Comparison of training with and without a value model. (a) Incorporating a value model reduces training fluctuations during training. This experiment is conducted on SDAR-4B-Chat. (b) A value model enables integration of trajectory-level process rewards, yielding faster optimization than relying solely on outcome rewards. This is conducted on SDAR-1.7B-Chat.

### 5.4 STRONGER OPTIMIZATION WITH TRACERL

We compare TraceRL with existing RL methods, focusing first on block diffusion models. Although current RL methods are primarily designed for full attention models, we adapt them directly to the block setting. For the random masking approach (Yang et al., 2025b), we restrict sampling within each block, making it resemble a semi-autoregressive method. For coupled RL (Gong et al., 2025), we introduce a complementary objective within each block, which yields more stable and effective training. We evaluate these methods on math tasks, with results shown in Figure 3. The experiments demonstrate that TraceRL achieves the best optimization performance, both with and without a value model. All RL methods use dynamic sampling during rollout, and TraceRL delivers superior optimization under both dynamic and static evaluation. These results highlight the importance of optimizing over the preferred trace, even within a small block.

### 5.5 ALSO FIT FOR FULL ATTENTION MODELS AND CODING TASKS

To demonstrate the broad applicability of TraceRL, we conduct experiments with full-attention models on coding RL tasks. For the adapted model, we employ Dream-7B-Coder-Instruct, first fine-tuning it on distillation data to provide a cold start and then performing RL training on stdio-format coding tasks (see Appendix D.5 for details). We use static sampling and set the shrinkage parameter to $s = 8$. For the pretrained DLM, we optimize LLaDA-8B-Instruct on functional-test coding tasks, applying dynamic sampling with $s = 1$. As in Table 1, we augment the training data for two baseline methods (Appendix D.5). We find that TraceRL converges faster and achieves the best performance compared with the baselines (see Figure 4). Notably, we achieve 25.0% accuracy on LiveCodeBench-V2, establishing a new state of the art among open-source full-attention DLMs.

### 5.6 DIFFUSION VALUE MODEL ENHANCES TRAINING STABILITY

Training a policy model coupled with a value model provides a variance-reducing baseline in LLM RL training (Schulman et al., 2017; Hu et al., 2025a). We extend this idea to diffusion language models. In our diffusion-based value model, the verifiable reward of the entire trajectory is assigned to the tokens in the final sampling step, while the reward for all preceding tokens is set to zero. We

Table 3: Adapting block size from $B = 4$ to $8$ on reasoning tasks with TraceRL. Reported values are accuracies of these baselines under dynamic sampling with threshold 0.9.

| Model | inference block size | MATH500 | GSM8K | LiveCodeBench |
|---|---|---|---|---|
| Base model with $B = 4$ | 4 | 67.4 | 88.9 | 11.2 |
| | 8 | 60.2 | 83.0 | 9.8 |
| Enlarge B to 8 by TraceRL | 8 | 67.7 | 88.7 | 10.8 |

then use our proposed modeling to estimate values and advantages, jointly optimizing the policy and value models. This approach reduces variance and stabilizes the training process (see Figure 5 (a)) for the 4B model on math tasks. Specifically, the variance of the reward during training drops from $6.6 \times 10^{-4}$ to $3.6 \times 10^{-4}$ (a reduction of about 45.5%). We also apply the same setting to SDAR-1.7B and find that the variance drops from $8.2 \times 10^{-4}$ to $5.3 \times 10^{-4}$ (a reduction of about 35.4%) when using the value model.

### 5.7 VALUE MODEL ACCOMMODATES PROCESS REWARD

Instead of assigning a single reward to the entire sequence, trajectory-level rewards enable more fine-grained supervision by incorporating process rewards for intermediate steps (Lightman et al., 2023). TraceRL leverages diffusion-based value model to integrate such process rewards. In this experiment, we use SDAR-1.7B-Chat as the base model for both the policy and the value networks, and employ Qwen3-4B as the process reward model, which provides rewards for intermediate segments of 200 tokens in addition to an overall verifiable reward (Appendix D.6). We set $(\gamma, \lambda) = (0.99, 1)$ in this experiment. Compared with using only the verifiable reward without a value model, incorporating process rewards yields more effective optimization (see Table 5 (b)).

### 5.8 SCALING BLOCK SIZE WITH TRACERL

A block size of 4 can limit inference flexibility, and some acceleration methods rely on larger block sizes (Hong et al., 2025; Song et al., 2025; Wang et al., 2025a). Therefore, we therefore explore using TraceRL to increase the block size. We first perform rollout with a block size of $B = 4$, then apply TraceRL with $B = 8$. After 60 steps, we switch the rollout to $B = 8$ and continue optimizing with $B = 8$ for 40 steps, yielding a model better adapted to larger block sizes. As shown in Table 3, this approach proves effective across different tasks, despite being trained only on math tasks.

### 5.9 LONG-CoT DIFFUSION LANGUAGE MODEL

The long-CoT diffusion language model TraDo-8B-Thinking is derived from TraDo-8B-Instruct. After obtaining TraDo-8B-Instruct through RL training, we further apply semi-autoregressive fine-tuning on 75K samples of OpenThoughts-3 dataset (Guha et al., 2025). As the first long-CoT diffusion language model, TraDo-8B-Thinking exhibits strong long-CoT reasoning capabilities across benchmarks (Table 7), achieving 85.8% accuracy on MATH500 (Table 2) and demonstrating that diffusion language models can also excel at complex reasoning tasks.

### 5.10 ANALYSIS ON ACCELERATION RATIO AND RESPONSE LENGTH

We observe that the acceleration ratio (Table 4) increases after TraceRL optimization. In particular, the 4B model achieves a $15.4\%$ speedup on MATH500. This can be explained by the model becomes more confident when encountering problems in the domain it has been optimized for, allowing each step of dynamic sampling more likely to unmask more tokens under the same threshold. We also observe that the average response length increases on most complex reasoning tasks.

### 5.11 ALTERNATIVE PARAMETERIZATIONS FOR VALUE MODEL TRAINING

We conduct ablation studies on the choices of $(\gamma, \lambda)$. The configuration $(\gamma, \lambda) = (1, 1)$ serves as the standard setting for value model training (Hu et al., 2025a). We additionally evaluate $(\gamma, \lambda) = (1, 0)$ and $(\gamma, \lambda) = (0.99, 1)$ using the SDAR-1.7B-Chat base model on math tasks with verifiable rewards under TraceRL. As shown in Table 5, the results remain robust across these parameterizations.

Table 4: Acceleration ratio and response length analysis. "Acceleration" is defined as the ratio of response length to the total sampling steps for a task under dynamic sampling, and we report the average ratio across the evaluation dataset. We use dynamic sampling with a threshold of 0.9 here.

| Model | MATH500 | | | LiveBench | | |
|---|---|---|---|---|---|---|
| | accelerated | total step | avg len. | accelerated | total step | avg len. |
| SDAR-4B-Chat | 2.28 | 240 | 548 tok. | 1.56 | 119 | 181 tok. |
| TraDo-4B-Instruct | 2.63 | 229 | 595 tok. | 1.61 | 154 | 238 tok. |
| SDAR-8B-Chat | 2.42 | 228 | 557 tok. | 1.56 | 161 | 256 tok. |
| TraDo-8B-Instruct | 2.50 | 240 | 625 tok. | 1.60 | 151 | 233 tok. |

Table 5: Accuracy on MATH500 for different $(\gamma, \lambda)$ after 60 training steps with the SDAR-1.7B

| | Base Model | (0.99, 1.0) | (1.0, 1.0) | (1.0, 0.0) |
|---|---|---|---|---|
| MATH500 Accuracy | 61.1 | 63.2 | 63.0 | 63.3 |

## 5.12 Ablation Study on Shrinkage Parameter

We evaluate different choices of the shrinkage parameter $s$. The RL experiments are conducted with the LLaDA-8B-Instruct base model on functional-test coding data. As shown in Table 6, smaller values of $s$ yield better performance, consistent with our finding that closely following the inference trajectory during optimization improves results. Nevertheless, larger values of $s$ still achieve strong performance relative to baseline methods. (Huang et al., 2025), as a special case with $s = 1$, can be seen as spending more training time to avoid any bias in training.

Table 6: Training time and accuracy for different $s$ after 60 training steps.

| | Base Model | s = 1 | s = 2 | s = 4 |
|---|---|---|---|---|
| MBPP Accuracy | 33.2 | 37.0 | 36.7 | 35.6 |
| A100 GPU hours / step | – | 3.7 | 2.6 | 2.1 |

## 5.13 Average Response Length of TraDo-8B-Thinking

We evaluate the average response length of TraDo-8B-Thinking across different evaluation datasets (Table 7). The results show that on more challenging tasks, the model produces longer responses with more reasoning steps and tokens. This observation is consistent with prior findings on long-CoT large language models (Guo et al., 2025; Hu et al., 2025a; Yang et al., 2025a).

Table 7: Average response length of TraDo-8B-Thinking on different benchmarks.

| | MATH500 | AIME2024 | GSM8K | LiveCodeBench | LiveBench |
|---|---|---|---|---|---|
| Avg. length | 5872 | 19397 | 1458 | 16291 | 16233 |

## 6 Conclusion

We present a new reinforcement learning method for diffusion language models with diverse architectures. Extensive experiments demonstrate the effectiveness of this method across multiple RL tasks, resulting in three state-of-the-art diffusion language models. We also highlight its benefits for accelerating inference and scaling block size, pointing to promising directions for future research.

# 7 ETHICS STATEMENT

Our research relies solely on publicly available benchmarks for math and coding tasks, and does not involve human subjects, private data, or applications with direct ethical risks.

# 8 REPRODUCIBILITY STATEMENT

The code is included in the supplementary material. We also describe the experimental details in Section 5.1 and Appendix D.

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

## APPENDIX CONTENTS

## A THEORETICAL RESULTS

### A.1 EXPLICIT EXPRESSIONS OF TOKEN-WISE RETURNS AND ADVANTAGES

**Proposition 1** (Token-wise return and advantage from step-wise recursions)**.** *Let a trajectory $\tau$ be partitioned into trace steps $\tau(1), \ldots, \tau(|\tau|)$, and let $t_j$ denote the unique step index with $j \in \tau(t_j)$. For token-wise rewards $r_j$ and token-wise values $V_j^{old}$, define the step-wise aggregates*

$$r_t^{\star} \; := \; \frac{1}{|\tau(t)|} \sum_{l \in \tau(t)} r_l, \qquad V_t^{\star,old} \; := \; \frac{1}{|\tau(t)|} \sum_{l \in \tau(t)} V_l^{old}.$$

*Let the step-wise return and GAE be given by*

$$R_t^{\star} \; = \; r_t^{\star} + \gamma R_{t+1}^{\star}, \quad R_{|\tau|+1}^{\star} = 0, \qquad \delta_t^{\star} \; = \; r_t^{\star} - V_t^{\star,old} + \gamma V_{t+1}^{\star,old},$$

$$A_t^{\star} \; = \; \sum_{k=0}^{|\tau|-t} (\gamma\lambda)^k \delta_{t+k}^{\star}, \quad A_{|\tau|+1}^{\star} = 0, \quad V_{|\tau|+1}^{\star,old} = 0.$$

*Define token-wise quantities*

$$R_j \; := \; r_j + \gamma R_{t_j+1}^{\star}, \qquad A_j \; := \; r_j - V_j^{old} + \gamma V_{t_j+1}^{\star,old} + \gamma\lambda A_{t_j+1}^{\star}.$$

*Then the following explicit expressions hold:*

$$R_j = r_j + \sum_{k=1}^{|\tau|-t_j} \gamma^k \frac{1}{|\tau(t_j+k)|} \sum_{l\in\tau(t_j+k)} r_l, \tag{1}$$

$$A_j = r_j - V_j^{old} + \sum_{k=1}^{|\tau|-t_j} (\gamma\lambda)^k \frac{1}{|\tau(t_j+k)|} \sum_{l\in\tau(t_j+k)} \left(r_l + \tfrac{1-\lambda}{\lambda} V_l^{old}\right). \tag{2}$$

*For the boundary case $\lambda = 0$, we have $A_j = r_j - V_j^{old} + \gamma V_{t_j+1}^{\star,old}$.*

**Proof.** **Return.** Employ $R_{t_j+1}^\star = \sum_{k=0}^{|\tau|-t_j-1} \gamma^k r_{t_j+1+k}^\star$ and substitute into $R_j = r_j + \gamma R_{t_j+1}^\star$ to obtain $R_j = r_j + \sum_{k=1}^{|\tau|-t_j} \gamma^k r_{t_j+k}^\star$. Using $r_t^\star = \frac{1}{|\tau(t)|}\sum_{l\in\tau(t)} r_l$ gives Equation 1.

**Advantage.** Write $A_{t_j+1}^\star = \sum_{k=0}^{|\tau|-t_j-1}(\gamma\lambda)^k\delta_{t_j+1+k}^\star$ and substitute $\delta_t^\star = r_t^\star - V_t^{\star,old} + \gamma V_{t+1}^{\star,old}$. Reindex to start at $k = 1$:

$$A_j = (r_j - V_j^{old}) + \sum_{k=1}^{|\tau|-t_j} (\gamma\lambda)^k r_{t_j+k}^\star + \gamma(1-\lambda)\sum_{k=1}^{|\tau|-t_j}(\gamma\lambda)^{k-1} V_{t_j+k}^{\star,old}.$$

The last line follows from collecting the $V^{\star,old}$ terms into a telescoping series whose coefficient is $\gamma(1-\lambda)(\gamma\lambda)^{k-1}$ for step $t_j + k$. Now use $r_t^\star = \frac{1}{|\tau(t)|}\sum_{l\in\tau(t)} r_l$ and $V_t^{\star,old} = \frac{1}{|\tau(t)|}\sum_{l\in\tau(t)} V_l^{old}$, and note that $\gamma(1-\lambda)(\gamma\lambda)^{k-1} = (\gamma\lambda)^k \cdot \frac{1-\lambda}{\lambda}$ for $\lambda > 0$. This yields Equation 2. For $\lambda = 0$, the series vanish and the definition gives the stated one-step TD form. $\qquad\square$

**Remark 1** (Some special cases). *(i)* $(\gamma, \lambda) = (1, 1)$. *We have*

$$R_j = r_j + \sum_{k=1}^{|\tau|-t_j} \frac{1}{|\tau(t_j+k)|} \sum_{l\in\tau(t_j+k)} r_l, \qquad A_j = R_j - V_j^{old},$$

*which is undiscounted Monte Carlo return and advantage with a token-wise baseline.*

*(ii)* $(\gamma, \lambda) = (1, 0)$. *The return remains the undiscounted form with $\gamma = 1$ as above, and*

$$A_j = r_j - V_j^{old} + V_{t_j+1}^{\star,old},$$

*which is the one-step TD advantage using the trace-level baseline $V_{t_j+1}^{\star,old}$.*

## A.2 ANALYSIS OF THE DEFAULT SETTING

We choose discount factor $\gamma = 1$ and TD parameter $\lambda = 1$ as the default setting, which is also commonly used in value models for LLMs (Hu et al., 2025a). We now analyze this choice of modeling diffusion-based value model in the following proposition.

**Proposition 2** (Analysis for the special case $(\gamma, \lambda) = (1, 1)$). *For simplicity, we denote $\mathcal{F}_t = \sigma(\tau(1{:}t))$. Given the token-wise reward $r_k$, we have the objective $J(\theta) = \mathbb{E}\left[\sum_{t=1}^{T}\sum_{k\in\tau(t)} r_k/|\tau(t)|\right]$. Define the step-wise score $g_t = \sum_{j\in\tau(t)} \nabla_\theta \log \pi_\theta(o_j \mid \mathcal{F}_{t-1})$. Then*

$$\nabla_\theta J(\theta) = \mathbb{E}\left[\sum_{t=1}^{T} g_t R_t^\star\right].$$

*For any prefix-measurable scalar baselines $b_t = b_t(\mathcal{F}_{t-1})$, the estimator of gradient is unbiased:*

$$\mathbb{E}\left[\sum_{t=1}^{T} g_t \left(R_t^\star - b_t\right)\right] = \nabla_\theta J(\theta). \tag{5}$$

*Moreover, define $\mu_t := \mathbb{E}[R_t^\star \mid \mathcal{F}_{t-1}]$, we have*

$$\mathbb{E}\left[\|g_t\|_2^2\left(R_t^\star - b_t\right)^2 \Big| \mathcal{F}_{t-1}\right] \le 2\,\mathbb{E}\left[\|g_t\|_2^2\left(R_t^\star - \mu_t\right)^2 \Big| \mathcal{F}_{t-1}\right] + 2\,\mathbb{E}\left[\|g_t\|_2^2 \mid \mathcal{F}_{t-1}\right]\cdot(\mu_t - b_t)^2. \tag{6}$$

*Proof.* Firstly, we have

$$\sum_{t=1}^{T} \frac{1}{|\tau(t)|} \sum_{k \in \tau(t)} r_k = \sum_{t=1}^{T} r_t^{\star}.$$

The trajectory density factorizes as $p_\theta(\tau) = \prod_{u=1}^{T} \prod_{k \in \tau(u)} \pi_\theta(o_k \mid \mathcal{F}_{u-1})$. Using the likelihood-ratio identity,

$$\nabla_\theta J = \int \Big( \sum_t r_t^{\star}(\tau) \Big) \nabla_\theta p_\theta(\tau) \, d\tau = \mathbb{E}\left[ \Big( \sum_t r_t^{\star} \Big) \sum_{u=1}^{T} \sum_{k \in \tau(u)} \nabla_\theta \log \pi_\theta(o_k \mid \mathcal{F}_{u-1}) \right],$$

Exchange the sums:

$$\nabla_\theta J = \mathbb{E}\left[ \sum_{u=1}^{T} \sum_{k \in \tau(u)} \Big( \sum_{t=1}^{T} r_t^{\star} \Big) \nabla_\theta \log \pi_\theta(o_k \mid \mathcal{F}_{u-1}) \right].$$

Fix $u$ and split the inner sum at $t < u$ and $t \geq u$. For $t < u$, $r_t^{\star} \in \mathcal{F}_{u-1}$, therefore

$$\mathbb{E}[r_t^{\star} \nabla_\theta \log \pi_\theta(o_k \mid \mathcal{F}_{u-1})] = \mathbb{E}\big[ r_t^{\star} \, \mathbb{E}[\nabla_\theta \log \pi_\theta(o_k \mid \mathcal{F}_{u-1}) \mid \mathcal{F}_{u-1}] \big] = 0,$$

since $\mathbb{E}[\nabla_\theta \log \pi_\theta(\cdot \mid \mathcal{F}_{u-1}) \mid \mathcal{F}_{u-1}] = 0$. Thus only $t \geq u$ remain:

$$\nabla_\theta J = \mathbb{E}\left[ \sum_{u=1}^{T} \sum_{k \in \tau(u)} \Big( \sum_{t=u}^{T} r_t^{\star} \Big) \nabla_\theta \log \pi_\theta(o_k \mid \mathcal{F}_{u-1}) \right].$$

Summing over $k \in \tau(u)$ gives $g_u$, yielding the stated identity. For the baseline, for any $b_u(\mathcal{F}_{u-1})$,

$$\mathbb{E}[g_u \, b_u] = \mathbb{E}[\mathbb{E}[g_u \mid \mathcal{F}_{u-1}] \, b_u] = \mathbb{E}\left[ \Big( \sum_{k \in \tau(u)} \mathbb{E}[\nabla_\theta \log \pi_\theta(o_k \mid \mathcal{F}_{u-1}) \mid \mathcal{F}_{u-1}] \Big) b_u \right] = 0,$$

so subtracting $b_u$ leaves the expectation unchanged, which completes the proof of Equation 5. Inequality 6 can be derived by Jensen inequality.

$\square$

**Remark 2** (Link to the policy surrogate with $\pi_\theta$ vs. $\pi_{\text{old}}$, and to the value loss). *(i) **Connection to the $\pi_\theta/\pi_{old}$ policy surrogate.** Evaluating the gradient identity of equation 5 at $\theta = \theta_{old}$ and using the standard importance ratio $r_j(\theta) := \pi_\theta(o_j \mid \mathcal{F}_{t-1})/\pi_{old}(o_j \mid \mathcal{F}_{t-1})$, the first-order surrogate whose gradient matches $\nabla_\theta J(\theta_{old})$ is*

$$\mathcal{L}_{\text{PG}}(\theta) \;=\; \mathbb{E}_{\tau \sim \pi_{old}}\left[ \sum_{t=1}^{T} \frac{1}{|\tau(t)|} \sum_{j \in \tau(t)} r_j(\theta) \big( R_t^{\star} - b_t \big) \right].$$

*Replacing $r_j(\theta)$ by the clipped ratio $C_\epsilon(r_j(\theta))$ yields the usual PPO-style surrogate, and adding a trust penalty ( $-\beta \, \text{KL}[\pi_\theta \| \pi_{old}]$ ).*

*(ii) **Connection between the value loss and the variance term.** Let the step-shared baseline be the token average $b_t = \frac{1}{|\tau(t)|} \sum_{j \in \tau(t)} V_j$, where each $V_j = V_j(\mathcal{F}_{t-1})$. With $\mu_t := \mathbb{E}[R_t^{\star} \mid \mathcal{F}_{t-1}]$ and the token-wise conditional targets $\mu_j := \mathbb{E}[R_j \mid \mathcal{F}_{t-1}]$, $j \in \tau(t)$, we have*

$$\mu_t - b_t \;=\; \frac{1}{|\tau(t)|} \sum_{j \in \tau(t)} \big( \mu_j - V_j \big),$$

*and by Jensen,*

$$(\mu_t - b_t)^2 \;\leq\; \frac{1}{|\tau(t)|} \sum_{j \in \tau(t)} \big( \mu_j - V_j \big)^2.$$

*Plugging this into the step variance term on the right-hand side of equation 6 shows that minimizing the token-level value regression loss $\mathcal{J}_{value} = \frac{1}{2} \mathbb{E}\big[ (V_{\theta_v}(\tau)_j - R_j)^2 \big]$ reduces an upper bound on $\mathbb{E}\big[ \|g_t\|_2^2 \, (R_t^{\star} - b_t)^2 \mid \mathcal{F}_{t-1} \big]$, thereby stabilizing the policy update.*

## B    RELATED WORK EXTENSION

### B.1    ENHANCING REASONING CAPABILITIES OF LARGE LANGUAGE MODELS

Approaches to enhance the reasoning abilities of large language models (LLMs), spanning Chain-of-Thought (CoT) prompting (Wei et al., 2022; Yang et al., 2024c)—which elicits step-by-step rationales—and self-improvement optimization algorithms (Zelikman et al., 2024; Hosseini et al., 2024; Yang et al., 2024d), in which the models iteratively refine their answers through verification, reflection, or voting. Incorporating Long-CoT methods, such as self-checking and self-reflection patterns, substantially improves the reasoning capabilities of LLMs. Reinforcement learning (RL) has also been shown to incentivize the reasoning ability of LLMs (Guo et al., 2025; Team et al., 2025; Hugging Face, 2025; Hu et al., 2025a) and to improve their ability to solve complex tasks (Jiang et al., 2025; Jin et al., 2025; Yang et al., 2024b; Wang et al., 2025c; Zou et al., 2025; Wang et al., 2025b; 2026).

### B.2    DIFFUSION MODELS FOR LANGUAGE MODELING

Extending the iterative-refinement paradigm of diffusion language models (Ho et al., 2020; Song & Ermon, 2019) to discrete language data is nontrivial. One line of research projects tokens into a continuous latent space where standard Gaussian diffusion can operate (Dieleman et al., 2022; Graves et al., 2023; Gulrajani & Hashimoto, 2023). A parallel line directly defines the forward noising process on the token simplex via state-transition matrices. Structured Diffusion LM (Austin et al., 2021) introduced a categorical forward kernel (e.g., uniform corruption or bit-flip) and trained a transformer to invert it. Subsequent work simplified or reparameterized the transition matrix to reduce variance and improve stability (Sahoo et al., 2024; Shi et al., 2024; Ou et al., 2024).

Masked Diffusion Models (MDMs) have emerged as a scalable architecture for large-scale diffusion language models (Nie et al., 2025; Ye et al., 2025; Yang et al., 2025b). By leveraging bidirectional attention, MDMs achieve stronger global consistency than left-to-right autoregressive transformers (Ye et al., 2024; Zhang et al., 2023), while their support for parallel decoding enables substantial inference acceleration (Labs et al., 2025; Google DeepMind, 2025).

### B.3    SAMPLING ACCELERATION

Although the parallel decoding strategy of diffusion language models theoretically promises faster sampling than LLMs, current open-source implementations remain slower, largely due to less mature infrastructure compared with optimized engines such as vLLM (Kwon et al., 2023). To address this, KV-cache techniques for diffusion language models have been proposed (Hu et al., 2025b; Liu et al., 2025; Ma et al., 2025; Arriola et al., 2025; Yu et al., 2025b; Wu et al., 2025), and several closed-source systems have already achieved faster sampling than LLMs (Labs et al., 2025; Google DeepMind, 2025; Song et al., 2025).

## C    ADDITIONAL EXPERIMENTAL RESULTS

### C.1    ADDITIONAL EXPERIMENTS ON DIFFERENT MODELS

To demonstrate the broad applicability of our method by extended experiments. For Dream, we first fine-tune its instruction model on our sampled 1K demonstration examples using Dream's official SFT setup, and then perform RL for 160 steps with a shrinkage value of 8. This yields a 9.4% accuracy improvement on math problems (from 49.2% to 58.6%). For LLaDA, we scale training on OpenCoder to 160 RL steps with a shrinkage value of 2. The OpenCoder dataset is filtered so that each remaining problem causes LLaDA to make at least one error given three attempts. Under this setting, we obtain 7.7% improvement on MBPP. For MMaDA, we train with RL on a curated dataset (including GeoQA, CLEVR, MMK12, and GQA) for 60 steps with a shrinkage value of 8. The training dataset is further filtered to remove all binary decision problems in order to avoid spurious reward signals. This improves MMaDA's visual reasoning ability, yielding a 5.1% gain on MathVerse (from 23.6% to 28.7%). For block diffusion models with different block sizes, we use SDAR-1.7B with block sizes of 4, 8, and 16, and run RL on the same math training set for 160 steps.

Table 8: Additional experiments on different models.

| Model | Task / Dataset | Before RL | After RL |
|---|---|---|---|
| Dream | MATH500 | 49.2 | 58.6 |
| LLaDA | MBPP | 33.5 | 41.2 |
| MMaDA | MathVerse | 23.6 | 28.7 |
| SDAR-1.7B-b4 | MATH500 | 55.9 | 60.9 |
| SDAR-1.7B-b8 | MATH500 | 46.5 | 52.8 |
| SDAR-1.7B-b16 | MATH500 | 45.0 | 52.4 |

We report the dynamic sampling accuracy before and after RL in Table 8, and observe consistent improvements on dynamic sampling across all block sizes.

## C.2 EXAMPLE OF LONG-COT OUTPUTS

### TraDo-8B-Thinking Example Output

**task:** What is the smallest positive perfect cube that can be written as the sum of three consecutive integers? **response:** Okay, so I need to find the smallest positive perfect cube that can be written as the sum of three consecutive integers. Let me start by understanding the problem step by step.

First, a perfect cube is a number that can be expressed as $n^3$ where n is a positive integer. For example, 1, 8, 27, 64, etc. The question is asking for the smallest one of these that can also be written as the sum of three consecutive integers.

Now, three consecutive integers can be represented as x, x+1, and x+2, right? So their sum would be x + (x+1) + (x+2). Let me compute that first.

Adding those together: x + x + 1 + x + 2 = 3x + 3. So the sum of three consecutive integers is 3x + 3, which can be factored as 3(x + 1).

Hmm, so the sum is always a multiple of 3. Therefore, any perfect cube that can be written as such a sum must be divisible by 3? Because it's three times some integer.

So, the cube must be divisible by 3. Let me note that down.

Therefore, the smallest perfect cube divisible by 3 is 27 (since $3^3$ = 27), but I need to check if 27 can be written as the sum of three consecutive integers. Let me check that.

Suppose 3x + 3 = 27. Then 3x = 24 =¿ x = 8. Therefore, the three numbers would be 8, 9, 10. Let's add them up: 8 + 9 + 10 = 27. Hey, that works! So 27 is indeed the sum of three consecutive integers.

But wait, hold on. The question says ïhe smallest positive perfect cube that can be written as the sum of three consecutive integers.Ïf 27 works, then is that the answer? But before I jump to conclusions, maybe there's a smaller cube that also satisfies this? Let me check.

The next smaller cube after 27 is 8 ($2^3$). Let's see if 8 can be written as the sum of three consecutive integers.

Setting up 3x + 3 = 8. Then, 3x = 5 =¿ x = 5/3, which is not an integer. Since x has to be an integer, this doesn't work. So 8 is out.

The next is 1 ($1^3$). Let's see, 3x + 3 = 1 =¿ 3x = -2 =¿ x = -2/3. Again, not an integer. So 1 is out.

Therefore, the next cube is 64 ($4^3$). Let's check that. Wait, but 64 is bigger than 27. Since 27 already works, but maybe there is another cube between 8 and 27? Since 8, 27, 64 are the cubes after 1, and 27 is the first one here.

Wait, hold on. The cubes are 1, 8, 27, 64, 125, etc. So between 8 and 27 is 27 which works. So 27 is the first one after 8. But does 27 itself work? Yes, as we saw.

But hold on, maybe the question is considering three consecutive integers in a different way? For example, maybe the integers can be negative? Let me check.

Suppose we allow negative integers. Let's see if there's a smaller cube that could be expressed as the sum of three consecutive (including negative) integers.

Let's check 1. 1 can't, as above. 8 can't. 27 can. 64? Let's see. Let me check 64.

3x + 3 = 64. 3x = 61 =¿ x = 20.333... Not integer. So no. 125? 3x + 3 = 125 =¿ 3x = 122 =¿ x = 40.666 nope.

So even with negative numbers, the next cube after 27 might still be 27. But let me think again. Maybe there's a different way to write three consecutive integers?

Wait, perhaps three consecutive numbers can be written as n-1, n, n+1? That would still be the same as x, x+1, x+2 where x = n-1. So yeah, same thing.

Alternatively, maybe three consecutive even or odd integers? But that would complicate the sum, but the problem didn't specify. The question just says three consecutive integers, so they can be any consecutive integers, whether positive or negative.

Alternatively, maybe I made a mistake in assuming that 27 is the first one. Let me confirm again.

The sum of three consecutive integers is 3x + 3 = 3(x + 1). So the cube must be divisible by 3.

Therefore, the cube must be a multiple of 3. The first cube divisible by 3 is 27 (3³). So yes, that must be the case.

Therefore, unless there is a smaller cube that is divisible by 3, but 1 is not, 8 is not, so 27 is the first one.

Therefore, the answer should be 27. But hold on, let me check with another approach to be sure.

Suppose the three consecutive integers are n, n, n+1. Wait, no. Three consecutive integers are n, n+1, n+2. So their sum is 3n + 3. So we need 3(n + 1) to be a perfect cube.

Therefore, $3(n + 1) = k^3$, where k is an integer. Therefore, $n + 1 = k^3 / 3$.

Since n must be an integer, $k^3$ must be divisible by 3. Therefore, k must be divisible by 3, because 3 is prime.

Therefore, let k = 3m, where m is an integer. Then:

$n + 1 = (27m^3)/3 = 9m^3$.

Therefore, $n = 9m^3 - 1$.

Therefore, the three integers would be $9m^3 - 1$, $9m^3$, $9m^3 + 1$. Their sum is $9m^3 - 1 + 9m^3 + 9m^3 + 1 = 27m^3$, which is indeed a cube.

Therefore, the cubes are all multiples of 27. Therefore, the smallest such cube is when m=1, which is $27*1^3 = 27$.

Therefore, 27 is indeed the smallest perfect cube that can be written as the sum of three consecutive integers.

Wait, but let me check m=0? If m=0, then n = -1, so the numbers would be -1, 0, 1. Their sum is 0, which is not a positive cube. So m=1 is the first positive case.

Therefore, this seems solid.

Therefore, the answer is 27.

But hold on, let me think again. The problem says s̈mallest positive perfect cube.S̈ince 27 is positive and a perfect cube, so it's the first one.

Alternatively, maybe there's a cube before 27 that can be expressed as the sum of three consecutive integers? Let me confirm again. The cubes are 1, 8, 27, 64, 125,...

Checking 1: 3x +3 =1: 3x= -2, nope.

8: 3x +3=8: 3x=5, nope.

27 works, so yeah, 27 is the first. Therefore, I think that's the answer.

Wait, but maybe I made a mistake in the equation. Let me recast the problem.

....... (We omit some reasoning steps here.)

Therefore, 27 is indeed the first.

Therefore, I think the answer is 27.

Wait, but the problem says s̈mallest positive perfect cube.: Since 27 is positive and perfect cube, and there are no smaller cubes that work, then yes, 27 is the answer.

Wait, but just to make sure, is 0 considered? But 0 is not positive. So 27 is the first.

Therefore, I think I can confidently say that the smallest positive perfect cube that can be written as the sum of three consecutive integers is 27.

**Final Answer**

27

# D EXPERIMENTAL DETAILS

## D.1 SUPERVISED FINETUNING METHODS EXPLORATIONS

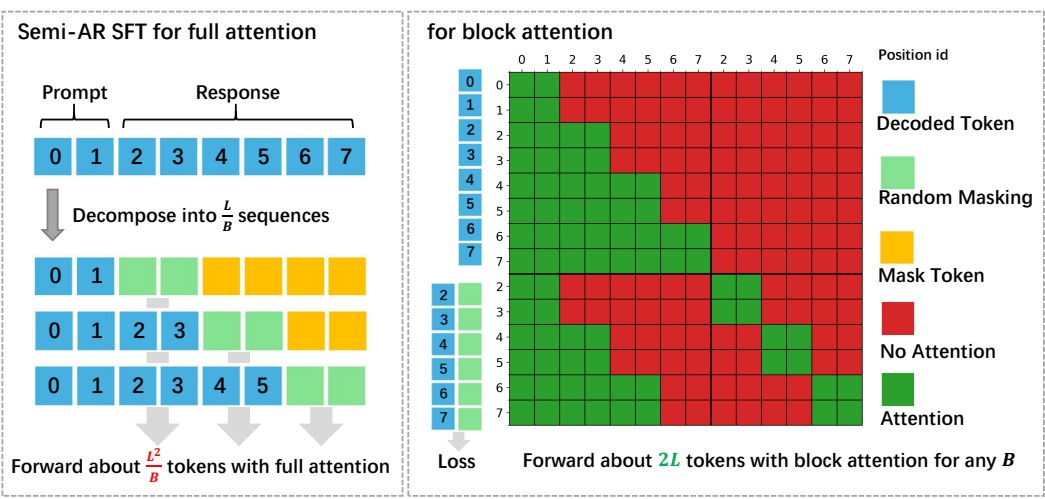

Figure 6: Semi-AR SFT for full attention and block attention model Overview. This example uses a block size of $B = 2$ and a sequence length of $L = 6$. Block diffusion models naturally adapt to semi-AR SFT efficiently, whereas full attention models require slicing the data by $B$.

In the demonstration of Section 3, the training data consist of CoT responses generated by the Qwen2.5-32B-Instruct model (Yang et al., 2024a) on 2,000 randomly selected tasks from the OpenR1-MATH training set (Hugging Face, 2025), filtered to exclude problems solvable by the Qwen2.5-7B-Instruct model. We then collect the traces of these data using the model under evaluation. For each instance, the model is used to iteratively select the two tokens with the highest confidence, conditioned on all previously processed tokens. This procedure produces a trace, which is then aggregated by grouping every $l/2$ neighboring tokens to obtain the final trace, where each step has length $l$. We train on the collected data for one epoch using 64 A100 GPUs, with a learning rate of $1 \times 10^{-6}$ for Dream and $1 \times 10^{-5}$ for SDAR. The semi-AR training objective is:

$$\mathcal{J}_{\text{semi}}(x, Q, \theta) = \sum_{i=1}^{\lceil L/B \rceil} \mathcal{J}_{\text{full}}\big(x^{(i-1)B:\min(iB,L)}, [Q, x^{0:(i-1)B}], \theta\big). \tag{7}$$

## D.2 PROMPT TEMPLATES

**Math Prompt Templates**

**Dream**

```
'''<|im_start|>system\nYou are a helpful assistant.<|im_end|>\n<|
    ↪ im_start|>user\nYou need to put your final answer in \\boxed
    ↪ {}. This is the problem:\n{{problem}}<|im_end|>\n<|im_start|>
    ↪ assistant\n'''
```

**LLaDA**

```
"""<|startoftext|><|start_header_id|>user<|end_header_id|>You need
    ↪ to put your final answer in \\boxed{}. This is the problem:\n
    ↪ {{problem}}<|eot_id|><|startoftext|><|start_header_id|>
    ↪ assistant<|end_header_id|>\n"""
```

**TraDo (non-thinking)**

```
'''<|im_start|>user\n{{problem}}\nPlease reason step by step, and
    ↪ put your final answer within \\boxed{}.<|im_end|>\n<|im_start
    ↪ |>assistant\n'''
```

**TraDo (thinking)**

```
'''<|im_start|>user\nYou need to put your final answer in \\boxed
    ↪ {}. This is the problem:\n{{problem}}<|im_end|>\n<|im_start|>
    ↪ assistant<think>\n'''
```

**TraDo (non-CoT)**

```
'''<|im_start|>user\nYou need to put your final answer in \\boxed
    ↪ {}. This is the problem:\n{{problem}}<|im_end|>\n<|im_start|>
    ↪ assistant\n'''
```

## Code Prompt Templates

**Dream**

```
'''<|im_start|>system\nYou are a helpful assistant.<|im_end|>\n<|
    ↪ im_start|>user\nThis is the problem:\n{{problem}}\nYou should
    ↪  put your code in '''python '''. Use input() to read input
    ↪ and print() to produce output in your script. <|im_end|>\n<|
    ↪ im_start|>assistant\n'''
```

**LLaDA**

```
'''<|startoftext|><|start_header_id|>user<|end_header_id|>This is
    ↪ the problem:\n{{problem}}\n You should put your code in '''
    ↪ python '''. Use input() to read input and print() to produce
    ↪ output in your script. <|eot_id|><|startoftext|><|
    ↪ start_header_id|>assistant<|end_header_id|>\n'''
```

**TraDo (non-thinking)**

```
'''<|im_start|>user\nThis is the problem:\n{{problem}}\nYou should
    ↪ put your code in '''python '''. Use input() to read input and
    ↪  print() to produce output in your script. <|im_end|>\n<|
    ↪ im_start|>assistant\n'''
```

**TraDo (thinking)**

```
'''<|im_start|>user\nThis is the problem:\n{{problem}}\nYou should
    ↪ put your code in '''python '''. Use input() to read input and
    ↪  print() to produce output in your script. <|im_end|>\n<|
    ↪ im_start|>assistant<think>\n'''
```

## Process Reward Prompt

**Process Reward (grading excerpt)**

```
'''<|im_start|>system\nYou are a helpful assistant.<|im_end|>\n<|
    ↪ im_start|>user\nFor the question below, you will be given an
    ↪ entire solution (we have already known it is {{if_correct}})
    ↪ and an excerpt (a middle part) from it. Your task is to grade
    ↪  the excerpt for correctness using 1 (correct) or -1 (
    ↪ incorrect).\n\nRules:\n- Judge ONLY the excerpt's correctness
    ↪  given the question and the whole solution (which is {{
    ↪ if_correct}}) as context.\n- The excerpt may be truncated at
    ↪ the beginning or end due to chunking; do NOT penalize
    ↪ boundary truncation.\n- If the excerpt contains any error,
    ↪ unjustified inference, or contradiction, score -1. If it is
    ↪ completely correct, score 1.\n- Ignore minor grammar issues
    ↪ that do not affect correctness.\n\nThis is the question:\n{{
    ↪ question}}\n\nThis is the final ground truth answer:\n{{
    ↪ gt_answer}}\n\nThis is the given solution ({{if_correct}}):\n
```

```
↪ {{whole_solution}}\n\nThis is the excerpt I need you to score
↪ :\n{{excerpt}}\n\nYou need to put your final score in \\boxed
↪ {}.\n<|im_end|>\n<|im_start|>assistant<think>\n'''
```

We use the same prompt template for both reinforcement learning and evaluation, with TraDo and SDAR models sharing identical templates. For long-CoT mode, we adopt the "thinking" prompt, while the non-CoT prompt is used only for SDAR in the toy experiment of Section 3, which evaluates the efficiency of different SFT methods in teaching the model to use CoT. Prompts are designed to align closely with model preferences and minimize formatting errors (e.g., failing to extract the final answer). For the process reward model, we set the temperature to 0.6 and the maximum generation length to 4096 tokens.

### D.3 EVALUATION DETAILS

For the LLaDA model, we use a block size of 32, a response limit of 1024 (with no output truncation observed), a temperature of 0.1, and a further horizon size of 128. The further horizon size is defined as the number of tokens forwarded beyond the target block, excluding the tokens within the block itself. We find that this additional window mitigates the performance drop caused by employing the KV-cache. For dynamic sampling, we set the unmasking threshold to $\mathcal{T} = 0.95$.

For the Dream model, we also use a temperature of 0.1, and a further horizon size of 128. For math problems, the response limit is set to 1600, and for coding problems, it is set to 1024. When using dynamic sampling, we use a block size of 4 with threshold $\mathcal{T} = 0.95$. Under static sampling, we use a block size of 32.

For the SDAR and TraDo instruction models, we keep the pretrained block size of 4, a response limit of 2000, and a temperature of 1.0. With dynamic sampling, we use a threshold of $\mathcal{T} = 0.9$ and $top\text{-}k = 0$ (i.e., all tokens are kept). For static sampling, we set $top\text{-}k = 1$, following (Cheng et al., 2025). For the long-CoT model TraDo-8B-Thinking, we set the response limit to 30,000 during evaluation, and only use dynamic sampling for evaluation.

For AIME2024, we evaluated each problem 20 times. For all other test datasets, we evaluated 3 times and report the average accuracy.

We use the KV-cache in all evaluations to accelerate inference. For full attention mask models, we adapt and improve the fast-dllm framework (Wu et al., 2025). For block diffusion models, we use jetengine (Cheng et al., 2025) to achieve acceleration.

### D.4 RL SAMPLING DETAILS

We report the parameters used in our RL experiments. For full-attention diffusion models, we set the block size to 32, the temperature to 0.8, and the further-horizon size to 128. At each step, we sample 56 problems with KV-Cache enabled, generating 8 responses per problem. For LLaDA, we adopt dynamic sampling to accelerate optimization with a response limit of 256. For the Dream model, we employ static decoding to improve sampling quality (Gong et al., 2025), with a response limit of 1024. For SDAR models, we use a block size of 4, dynamic decoding with threshold $\mathcal{T} = 0.9$, $top\text{-}k = 0$, temperature 1.0, and $top\text{-}p = 1.0$ (also applied during evaluation). At each step, we sample 128 problems and generate 32 responses per problem.

### D.5 TRAINING DETAILS

For full attention models, we use a learning rate of $1 \times 10^{-6}$ for both semi-autoregressive and fully random masking fine-tuning methods. For block attention models, we use a learning rate of $1 \times 10^{-6}$. The masking probability is uniformly sampled between 0.1 and 0.9. Before RL training in figure 4(a), we collect 1.7k random SFT training samples from CodeContest (Li et al., 2022), with solutions generated by the Qwen2.5-32B-Instruct model. We find that including the eos token in the SFT data is beneficial for stabilizing RL training.

During RL training, we set the learning rate to $1 \times 10^{-6}$, with $\beta = 0.01$ and $\epsilon = 0.2$. When using a value model, we find no significant difference between the common parameter choices (Section C). By default, we use the $k = 3$ estimator for KL. For math tasks, we use binary outcomes as verifiable rewards and retain only those tasks with accuracy between 0.2 and 0.8 for training (Yu et al., 2025a). For coding tasks, we use as the reward the proportion of unit tests passed by the generated solutions. To accelerate training, we use 64 A100 GPUs for our demonstration and ablation experiments. However, all experiments can be conducted on 8 A100 GPUs.

In figure 4 (a), for Trace RL we use a shrinkage parameter of $s = 8$, together with an average response length of 380 during the RL process, resulting in approximately $380/8 = 47.5 < 50$ forward passes per data point. We augment the training data for fully random masking and coupled methods by applying 25 independent random masks, yielding 50 training samples per data point for the coupled method and 25 training samples per data point for the random masking method. Following Gong et al. (2025), we keep the number of training samples for random masking at half that of the coupled method. Similarly, in Figure 4(b), the average response length is 60 and the average acceleration ratio (see Table 4) is 2.0. Accordingly, we augment the training data for the coupled and fully random masking methods by a factor of 15.

One noteworthy point is the number of pad tokens that need to be trained for each data point, which we denote as $n_{pad}$. Setting a large $n_{pad}$ leads to excessively large logits for the pad token and can cause the model to terminate inference prematurely. In our RL training, we set $n_{pad} = 0$, which works well. However, in our long-CoT SFT step, we find that setting $n_{pad} = 0$ can potentially cause the model to never stop generating output during inference, although one remedy is to add an eos token as the stop token. Therefore, choosing an appropriate $n_{pad}$ is vital for stable training.

## D.6 PROCESS REWARD MODELING

In our experiments on trajectory process rewards, we use Qwen3-4B to assign rewards for each 200-token segment. Specifically, we divide the model's response into excerpts of length 200 and query Qwen3-4B three independent times using the prompt described in Section D.2, with each query producing a binary score ($+1$ or $-1$). If all three evaluations agree, we adopt the majority score as the reward; otherwise, we fall back to the verifiable reward ($+1$ or $-1$) for that segment. After assigning rewards, we normalize them per problem and per segment: for each problem and segment index (e.g., the first 200-token segment), we pool rewards from all generated responses and normalize that group jointly; segments are normalized independently. The normalized rewards are assigned to the tokens in the last step within each segment.

### D.7 TraceRL Algorithm Pipeline

Algorithm 1: TraceRL (Trajectory-Aware RL for DLMs)

1: **Input:**
2:     1) Task set $\mathcal{D}_{\text{task}} = \{Q_1, \ldots, Q_N\}$.
3:     2) Policy $\pi_{\theta_p}$ parameterized by $\theta_p$.
4:     3) (Optional) Value network $V_{\theta_v}$ parameterized by $\theta_v$; flag `UseValue` $\in \{\text{True}, \text{False}\}$.

5:     4) Iterations $M$, rollouts per task $G$, PPO clip $\epsilon$, KL coefficient $\beta$.
6:     5) Discount $\gamma$, GAE parameter $\lambda$, learning rates $(\eta_p, \eta_v)$.
7:     6) Shrinkage parameter $s$ (aggregate every $s$ neighboring trace steps).
8:     7) (Optional) Value update interval $E_v$ (update value every $E_v$ iterations).
9: **Initialize:** $\theta_p$ (and $\theta_v$ if `UseValue`).
10: **for** $t = 1$ to $M$ **or** not converged **do**
11:     $\pi_{\text{old}} \leftarrow \pi_{\theta_p}$     *// freeze old policy for PPO ratios & KL*
12:     **if** `UseValue` **then**
13:         $V_{\text{old}} \leftarrow$ freeze copy of $V_{\theta_v}$     *// stop-gradient baseline*
14:     **end if**
15:     **Sample rollouts (trajectory traces):**
16:     **for each** minibatch of tasks $Q \sim \mathcal{D}_{\text{task}}$ **do**
17:         **for repeat** $G$ times **do**
18:             Generate a response by policy decoding to obtain a trace $\tau = (\tau(1), \ldots, \tau(|\tau|))$, where $\tau(t)$ is the token set decoded at step $t$.
19:             Obtain verifiable reward $r$ (or optionally process-level token/step rewards).
20:             **Shrink the trace:** $\tau^s \leftarrow \text{shrink}(\tau, s)$, i.e., group every $s$ neighboring steps.
21:             **if** `UseValue` **then**
22:                 **Value inference:** $V_j^{\text{old}} \leftarrow (V_{\text{old}}(\tau))_j$ for $j \in \tau$;    $V_t^{\star,\text{old}} \leftarrow \frac{1}{|\tau(t)|} \sum_{j \in \tau(t)} V_j^{\text{old}}$.

23:                 **Build returns/advantages on the trace:**
24:                   Calculate step-wise reward: $r_t^\star$, return $R_t^\star$, advantage $A_t^\star$.
25:                   Map to tokens ($j \in \tau(t)$): $R_j \leftarrow r_j + \gamma R_{t+1}^\star$,    $A_j \leftarrow (r_j - V_j^{\text{old}}) + \gamma V_{t+1}^{\star,\text{old}} + \gamma\lambda A_{t+1}^\star$.
26:                   (Optional) normalize advantages: $A_j \leftarrow \text{normalize}(A_j)$.
27:             **end if**
28:             Store $(Q, \tau^s, \text{advantages}, \pi_{\text{old}})$ for policy update.
29:         **end for**
30:     **end for**
31:     **Build grouped dataset** $\mathcal{D}_{\text{grp}} \leftarrow \{(\mathbf{p}_g, \mathcal{O}_g, A_g, \pi_{\text{old}})\}$.
32:     **for** $e = 1$ to $K$ **do**
33:         Sample minibatch $\mathcal{M} \subset \mathcal{D}_{\text{grp}}$; compute ratios $r_j = \frac{\pi_{\theta_p}(o_j | \mathbf{p}_g)}{\pi_{\text{old}}(o_j | \mathbf{p}_g)}$.
34:         **Policy step:** maximize $\mathcal{J}_{\text{policy}}(\theta_p)$ (Eq. equation 3);    $\theta_p \leftarrow \theta_p + \eta_p \nabla_{\theta_p} \mathcal{J}_{\text{policy}}$.
35:         **if** `UseValue` **and** ($t \bmod E_v = 0$) **then**
36:             **Value step:** minimize $\mathcal{J}_{\text{value}}(\theta_v)$ (Eq. equation 4);    $\theta_v \leftarrow \theta_v - \eta_v \nabla_{\theta_v} \mathcal{J}_{\text{value}}$.
37:         **end if**
38:     **end for**
39: **end for**
40: **Output:** Trained policy $\pi_{\theta_p}$ (and value $V_{\theta_v}$ if used).

## E USE OF LARGE LANGUAGE MODELS

We use large language models solely to polish English writing and improve clarity. All research ideas, methods, and results are entirely our own. All experiments involving large language models and diffusion language models were conducted in accordance with the statements of this paper.

