# OpenReview forum: "Revolutionizing Reinforcement Learning Framework for Diffusion Large Language Models"
_ICLR.cc/2026/Conference — ICLR 2026 Poster_

### Official Review · Reviewer_bxFJ · 2025-10-28

**Soundness:** 3
**Presentation:** 3
**Contribution:** 3
**Rating:** 6
**Confidence:** 4

**Summary:**

This paper studies reinforcement learning for diffusion language models. During the learning of diffusion models, one needs to sample noised intermediate samples to learn reconstructions. The authors highlight the importance of using noisy samples from diffusion models' sampling trajectories in training. They also introduce using value model to enhance training stability. The effects are validated on math and coding benchmarks, which surpasses strong autoregressive instruct models.

**Strengths:**

1. Enabling the use of reinforcement learning is an crucial steps to advance diffusion language models. This paper provides timely and systematic investigation into this topics.
2. This paper validates the effect of several intuitive techniques in RL for diffusion language models, which is comprehensive and helpful for following research, including shrinkage steps, aligning training and sampling trajectories, introducing value models, and sliced training.
3. The results is superior, matching strong autoregressive language models like Qwen2.5-7B-Instruct on representative math and coding benchmark. As far as I know, this is the first RL diffusion language models to achieve such results on these convincing benchmarks.
4. To the best of my knowledge, this paper also implement the first diffusion language models capable of long-cot thinking.

**Weaknesses:**

1. The comparison to autoregressive models is not strictly fair. Strong results are only demonstrated through SDAR, which is a block diffusion language models with extremely small block sizes. This does not convince me on the potential of improving a diffusion language models. And the long-cot models are not compared to long-cot autoregressive models.
2. The results do not support the claims about benefits of introducing value models. In figure 3, the results with value are almost identical to the ones without value model.

**Questions:**

1. Could you show more results on full attention diffusion language models without blocks (i.e., LLaDA, Dream instead of SDAR) ? How about the effect of RL on diffusion language models given different block sizes?

2. How long is the responses of TarDo-Instruct compared to Qwen2.5-Instruct?

---

> ### Author Response · Authors · 2025-11-18
> **Response to Reviewer bxFJ (part 1/2)**
>
> Thank you for your appreciation of this work and for your detailed comments!
>
> **Q1: Strong results are only demonstrated through SDAR.**
>
> 1. In our paper, Dream-Coder after RL achieves 25.0% accuracy on LCB-v2, which is the **state-of-the-art performance among all open-source full-attention DLMs.** To make our results more convincing, we further **extend our experimental settings in Response 2**, demonstrating the significance of TraceRL across more settings.
> 2. Secondly, the reason that only RL on the SDAR-based model surpasses LLMs such as Qwen2.5-Instruct is that the final performance is largely determined by the quality of the pretrained backbone. **Neither LLaDA nor Dream shares the same high quality training data as Qwen, so it is natural that, even after RL, they may still fall short of Qwen.**
>
> **Q2: Show more results on full attention DLMs, and block DLMs with different block sizes?**
>
> Of course! To demonstrate the broad applicability of our method, we add experiments on Dream, LLaDA, and MMaDA for math, coding, and vision–reasoning tasks, respectively, and also conduct experiments on block diffusion models with different block sizes (SDAR-1.7B-b4 / b8 / b16) on math tasks. **All results have now been updated in the Appendix of our paper (blue text).**
> 1. For Dream, we first fine-tune its instruction model on our sampled 1K demonstration examples using Dream’s official SFT setup, and then perform RL for 160 steps with a shrinkage value of 8. We obtain a 9.4% accuracy improvement, demonstrating that RL on full-attention DLMs can also yield significant gains.
>
> **Table 1. RL for Dream on math problems**
> ||Acc|
> |-|-|
> |Before RL|49.2|
> |After RL|58.6 (*+9.4*)|
>
> 2. For LLaDA, we scale training on OpenCoder to 160 RL steps with a shrinkage value of 2. The OpenCoder dataset is filtered so that each remaining problem causes LLaDA to make at least one error given three attempts. From the following table. we achieve a significant 7.7% improvement on MBPP.
>
> **Table 2. Scale LLaDA on MBPP**
> ||Acc|
> |-|-|
> |Before RL|33.5|
> |After RL|41.2 (*+7.7*)|
>
> 3. For MMaDA, we train with RL on a curated dataset (including GeoQA, CLEVR, MMK12, and GQA) for 60 steps with a shrinkage value of 8. The training dataset is further filtered to remove all binary decision problems in order to avoid spurious reward signals. This improves MMaDA’s visual reasoning ability, yielding a 5.1% gain on MathVerse.
>
> **Table 3. RL for MMaDA on visual reasoning**
> ||Acc|
> |-|-|
> |Before RL|23.6|
> |After RL|28.7 (*+5.1*)|
>
> 4. For block diffusion models with different block sizes, we use SDAR-1.7B with block sizes of 4, 8, and 16, and run RL on the same math training set for 160 steps. We report the dynamic sampling accuracy of these models on MATH500 in the following table, and observe that all block sizes achieve significant improvements.
>
> **Table 4. RL for block diffusion models with different block sizes**
> |block size|Before RL|After RL|+Acc|
> |-|-|-|-|
> |4|55.9|60.9|+5.0|
> |8|46.5|52.8|+6.3|
> |16|45.0|52.4|+7.4|

---

> ### Author Response · Authors · 2025-11-18
> **Response to Reviewer bxFJ (part 2/2)**
>
> **Q3: In figure 3, can't see any difference, the results with value are almost identical to the ones without value model.**
>
> To make the difference more distinguishable, we add **quantitative analysis**. In Figure 3, the variance of the reward during training drops from 6.6e-4 to 3.6e-4 (**a reduction of about 45.5%**). We also apply RL to SDAR-1.7B and find that the variance drops from 8.2e-4 to 5.3e-4 (**a reduction of about 35.4%**) when using value model. These numerical results demonstrates the significance of our diffusion value model. We have updated these numerical results in the paper.
>
> **Q4: long-cot models are not compared to long-cot autoregressive models.**
>
> 1. Our long-CoT models are fine-tuned on only 75K examples from the OpenThoughts-3 dataset (1/16 of the full dataset) due to resource limitations, achieving a 23.7\% improvement on AIME2024. For a fair comparison, we fine-tune Qwen2.5-7B-Instruct on the same data, which yields a 25.4\% improvement on AIME2024, showing that the gains are comparable.
> 2. One of the reason that TraDo-Thinking can not outperform SoTA 8B scale reasoning model, like Qwen3, is the scale of post training data they have upon us. However, the potential of DLM is still worth exploring due to its advantage in parallel acceleration (Gemini Diffusion, D2F), unified modeling (MMaDA, Lumina), iterative refinement (Edit Flows), and super data learner [2].
>
> [2]. Diffusion Language Models are Super Data Learners
>
> **Q5: How long is the responses of TarDo-Instruct compared to Qwen2.5-Instruct?**
>
> From the following table (where "Before RL" is SDAR-8B and "After RL" is TraDo-8B), we observe that the response length of our DLM (TraDo-8B) surpasses that of the Qwen instruction model, providing further evidence of its stronger reasoning ability.
>
> **Table 5. Response length on MATH500**
> |Model|Length|
> |-|-|
> |Before RL|557|
> |Qwen2.5-7B-Instruct|582|
> |After RL|625|
>
> Please feel free to raise any additional concerns. If you find that your main issues have been resolved, we would sincerely appreciate it if you could consider increasing the score.

---

### Official Review · Reviewer_hx5e · 2025-10-28

**Soundness:** 3
**Presentation:** 3
**Contribution:** 3
**Rating:** 8
**Confidence:** 5

**Summary:**

The paper introduces TraceRL, a trajectory-aware RL framework for diffusion language models (DLMs). It argues that standard post-training objectives for DLMs (fully random masking; semi-autoregressive SFT) are misaligned with the left-to-right inference trajectories used at decoding time, and proposes to (i) aggregate decoding steps into short “trace” segments via a shrinkage parameter , (ii) optimize a PPO-style policy objective over these segments, and (iii) pair the policy with a diffusion-style value model that provides variance-reduced advantages and supports process rewards. The approach improves math/coding benchmarks and can even adapt models to larger block sizes

**Strengths:**

1. The paper carefully motivates why training with random masking or generic semi-AR losses can be misaligned with the actual iterative unmasking/decoding procedure (static vs dynamic sampling), and shows a simple analysis/experiment to support this.

2. The algorithm aggregates 𝑠 consecutive decoding steps into segments, computes advantages/returns at the segment (and token) level, and updates a PPO-style objective. This aligns learning signals with how the model is actually used at inference. The paper gives explicit value loss (clipped regression) and describes sliced training under block diffusion to keep training efficient.

3. Extending variance-reduction baselines to DLMs is novel here, with the verifiable reward assigned at the final sampling step and zeros elsewhere; the same machinery accommodates mid-trajectory “process” rewards. Empirically this stabilizes training and accelerates optimization.

4. On math/coding, TraceRL improves over strong DLM baselines; the TraDo models achieve the best results among compared DLMs, with detailed tables.

**Weaknesses:**

1. For the value model, the entire verifiable reward is assigned to tokens in the last sampling step, with zeros elsewhere. This is simple but may bias learning toward late segments and under-credit earlier reasoning。

2. Process rewards come from Qwen3-4B judging 200-token segments with a fallback to the verifiable reward. The paper should quantify judge agreement rates, false-positive penalties (reward hacking) and robustness to prompt variations.

3. While the paper compares to several full- and block-attention DLMs, it would help to include stronger autoregressive SOTA (e.g., top open 7–8B instruct models) under matched decoding/training budgets.

4. The paper provides explicit forms for token-wise advantages and a clipped value loss, but there’s no convergence or bias/variance analysis for the “trace shrinkage” estimator or the “sliced” block training.

**Questions:**

1. The authors are expected to provide an ablation on reward assignment (last-step vs. proportional attribution) and an analysis of how attribution interacts with acceleration ratios/response lengths.

2. It would be better if the authors could give judge prompts/configs and inter-rater stats; run sanity checks (shuffled/adversarial segments) to quantify false positives.

3. The authors are expected to include strong AR instruction-tuned models under matched decoding/compute

---

> ### Author Response · Authors · 2025-11-18
> **Response to Reviewer hx5e (part 1/2)**
>
> Thank you for your appreciation of our work and for your insightful comments!
>
> **Q1. For value model, verifiable reward is assigned to tokens in last sampling step, with zeros elsewhere. Simple but may bias learning. Provide an ablation (proportional attribution) and analyze how attribution interacts with acceleration ratios/response lengths.**
>
> 1. Although the verifiable reward is assigned only to the tokens at the last step, setting $\gamma = 1$ will attribute this reward equally to all previous steps through the return (see Proposition 1 in the paper). If we instead directly assign the reward equally to every position, the first few steps will accumulate extremely large return values (proportional to the response length), which leads to the instability observed in our experiments.
> 2. However, we can still study how the discount factor $\gamma$ affects the attribution of returns. The smaller $\gamma$ is, the smaller the proportion of reward signal that earlier steps receive. In the ablation studies in our paper, we evaluated performance under different values of $\gamma$; here, we further extend this analysis to acceleration ratio and response length, experimenting on SDAR-4B. Based on the results below, we find that all metrics are robust to variations in $\gamma$.
>
> **Table 2. Ablation on $\gamma$**
> |$\gamma$|acceleration ratio|response length|
> |-|-|-|
> |1.0|2.63|595|
> |0.99|2.60|589|
> |0.9|2.65|602|
>
> **Q2: Quantify judge agreement rates, give judge prompts/configs and inter-rater stats, sanity checks (shuffled/adversarial segments) to quantify false positives.**
>
> A2: We add the following experiments to further study the robustness of our process reward modeling.
>
> 1. First, judge agreement rate: for SDAR-4B’s evaluation results on MATH500, we use Qwen3-4B and Qwen3-8B as two independent judges. We then compute their average agreement over all chunks and obtain a value of 0.86, which demonstrates the robustness of using the Qwen3 reasoning model as the process reward model.
> 2. Second, we perform an ablation on our judging prompts. Our original judge prompts are provided in Appendix D.2. The temperature is set to 0.6 and the maximum generation length to 4096 tokens, which we have now clarified in the paper (Appendix D.2). We then change the main scoring rule to: if the excerpt contains logical errors or insufficient reasoning (e.g., skipped steps), we assign a score of −1; if it is correct and sufficient, we assign a score of 1. We rerun the experiment in Figure 5 under this new rule. The results in Table 2 show that our conclusions are robust to the choice of prompting.
>
> **Table 2. Acc using different reward prompts.**
> |Prompts|acc|
> |-|-|
> |Original|63.0|
> |New|63.2|
>
> 3. Finally, we evaluate the false positives of process rewarding by deliberately corrupting the evaluation results of SDAR-4B on MATH500: (1) inserting garbled text and (2) shuffling some words (evenly) in a subset of chunks. We find that Qwen3-4B and Qwen3-8B can detect these errors with probabilities of 0.95 and 0.98, respectively, which demonstrates the robustness of our prompting on reasoning process reward models.
>
> **Q3: Include strong AR instruction-tuned models under matched decoding/training.**
>
> In our long-CoT SFT experiment, we use 75K examples from the OpenThoughts-3 dataset (1/16 of the full set) for SFT, achieving a 23.7% improvement on AIME2024. For a fair comparison, we fine-tune Qwen2.5-7B-Instruct on the same data, which yields a 25.4% improvement on AIME2024, showing that the gains are comparable.

---

> ### Author Response · Authors · 2025-11-18
> **Response to Reviewer hx5e (part 2/2)**
>
> **Q4: Bias/variance analysis for the trace shrinkage and sliced block training.**
>
> 1. For the shrinkage parameter, we extend our paper’s ablation study by training RL LLaDA on OpenCoder with different shrinkage values s, evaluated on MBPP. From the results, we observe that larger s introduces bias but accelerates training. This trade-off allows users to choose an appropriate s based on their requirements.
>
> **Table 3. ablation on s, for performance and efficiency**
> |s|acc|GPU time per step|
> |-|-|-|
> |1|37.0 (+3.8)|3.7 A100 GPU hours|
> |2|36.7 (+3.5)|2.6 A100 GPU hours|
> |4|35.6 (+2.4)|2.1 A100 GPU hours|
>
> 2. Sliced block training does not introduce bias due to the mechanism of block attention: the forward computation of each block depends only on the previous blocks and the current block, and not on any later blocks. This allows us to safely perform sliced block training while simultaneously accelerating training.

---

### Official Review · Reviewer_YmJb · 2025-10-31

**Soundness:** 2
**Presentation:** 3
**Contribution:** 3
**Rating:** 4
**Confidence:** 4

**Summary:**

This paper studies RL methods to reinforce the whole multi-step generation trajectories in DLLM -- this can be applied to both full-attention version and block-attention version. To accelerate training, the actual denoising steps are grouped into fewer actions to reinforce. Moreover, the paper introduce a step-wise value model to estimate an intermediate result to reduce the variance during. The experiments are conducted on TraDo, and achieve higher performance on math and code benchmarks.

**Strengths:**

1. This paper provides a method to shrink the trajectory length to reinforce, accelerating training
2. Introducing value model in DLM RL algorithm is novel, and it is reasonable to evaluate the quality of an intermediate steps.
3. The paper is well written and easy to follow.

**Weaknesses:**

1. Why do we only shrink the steps after sampling trajectory? Actually, we can also just sample shorter trajectories by allowing more tokens to be predicted in each step.
2. The paper does not discuss the effect of shrinking $s$ consecutive steps, both on the final performance and the training efficiency. [1] reinforces the whole trajectory without shrinking. It worth a comparison.
3. The pretrained model in L178 is actually Qwen as in Appendix D.1, right? It is quite confusing and misleading, since usually such a phrase means the model before post training. Why not sampled from the DLM itself, just like the coldstart phase in Deepseek-R1?


[1] Reinforcing the Diffusion Chain of Lateral Thought with Diffusion Language Models

**Questions:**

1. How to calculate advantage in the "w/o value model" setting?
2. If the preference inference traces are all left-to-right ones from Qwen, what is the final generation order preference after our RL training?
3. Since intermediate steps may contain mask tokens, how can Qwen3-4B be used as a process reward model to evaluate such intermediate outputs?

---

> ### Author Response · Authors · 2025-11-18
> **Response to Reviewer YmJb**
>
> Thank you for your detailed comments for our paper!
>
> **Q1: Why only shrink the steps after sampling? Allow more tokens predicted each step?**
>
> Because sampling more tokens (e.g., use higher threshold) leads to degradation in generation quality (Table 1), generating too many tokens in a single step causes overconfidence, such as duplicated, missing, or misplaced tokens, which limits RL’s efficiency in exploring high-quality samples. **Shrinking keeps the samples high quality while accelerating training.**
>
> **Table 1. Acc vs threshold for SDAR-4B.**
> |sampling|acc|
> |-|-|
> |static, 1 token/step|70.2|
> |static, 2 tokens/step|54.8|
> |dyn, 0.9|67.4|
> |dyn, 0.8|65.5|
> |dyn, 0.7|64.8|
> |dyn, 0.6|62.8|
>
> **Q2: [1] worth a comparison. Not discuss the effect of $s$, both in performance and efficiency.**
>
> Thanks for your reminder! [1] is a great work. The comparison of different cases (including \(s = 1\)) is in Appendix C.2. **We have now revised the paper to add discussion of [1] in both the related work (section 2.3) and Appendix C.2 (in blue text).** **In the paper, we had already conducted an ablation on the effect of \(s\) on performance, but we initially omitted the training efficiency**; we now report the training efficiency as well (see following and Appendix C.2). We find there is a trade-off between quality and efficiency: smaller \(s\) makes the less bias, but also slows down training.
>
> **Table 3. ablation on s, for performance and efficiency**
> |s|acc|GPU time per step|
> |-|-|-|
> |1|37.0 (+3.8)|3.7 A100 GPU hours|
> |2|36.7 (+3.5)|2.6 A100 GPU hours|
> |4|35.6 (+2.4)|2.1 A100 GPU hours|
>
> **Q3: Pretrained model in L178 is Qwen?**
>
> The pretrained model mentioned in L179 refers to the model that collects the trajectories; **it is simply the DLM to be SFT, not Qwen**. The SFT data are sampled by Qwen, but **the trajectory information (i.e., which step each token belongs to) is collected by the DLM to be SFT (see Appendix D.1 for details)**. In other words, the same data (generated by Qwen) can induce different preference trajectories for different DLMs. We clarify this in current version to avoid any confusion.
>
> **Q4: Why not sampled from the DLM itself in SFT demonstration experiment (like R1)?**
>
> 1. We choose a fixed SFT dataset (generated by Qwen) so that the demonstration experiments are consistent across different DLMs. Second, SFT data needs to be exceptionally high quality to distinguish the effects of different SFT methods. If we directly sampled from a DLM and then performed SFT on itself, this would be almost equivalent to performing a single RL step, which is not suitable for a demonstration experiment because a single RL step does not yield a clearly distinguishable effect. Therefore, we filter and distill high-quality demonstration data from strong Qwen models.
> 2. Our main RL results show that, after hundreds of RL steps, the advantage of the trajectory-aware method becomes significant. The SFT demonstration experiment, in contrast, is designed to help readers better understand the core idea, rather than to serve as a cold-start recipe like R1. Our RL experiments are applied directly to instruction models without any additional SFT.
>
> **Q5: How to calculate advantage in the "w/o value model" setting?**
>
> 1. Without a value model: we directly use the standardized reward as the advantage for each response (see Section 4.1). In this setting, the advantage is identical at every token position.
> 2. With a value model, we first assign a reward to each token position and then apply our diffusion value model to compute the value at each token position (step-wise). Given these two quantities, we derive the advantage at each token position using the algorithm described in Proposition 1 of our paper (see also Algorithm 1).
>
> **The process of calculating/training value/return/advantage is in Section 4.2 and Proposition 1 in our paper.**
>
> **Q6: If preference inference traces are all left-to-right ones from Qwen, what is the final generation order preference after our RL training?**
>
> **The preference inference traces are not obtained from Qwen, but from the DLMs themselves (see Response 3).** In addition, we do not perform SFT before RL training, so the RL procedure purely optimizes over each DLM’s own trajectories.
>
> **Q7: Intermediate steps may contain mask tokens, how can process reward model evaluate such outputs?**
>
> To ensure that no mask tokens are involved, we set the chunk length (used to divide the response into chunks and assign process reward) to be a multiple of the block size (see Fast-dLLM, where the block size is used to perform semi-AR high-quality sampling with KV cache). This block-wise semi-AR inference guarantees that the process reward model never receives any mask tokens and instead focuses on evaluating intermediate reasoning steps that are already fully unmasked.
>
> Given that all misunderstanding has been clarified, we would greatly appreciate it if the reviewer could consider increasing the score.

---

> > ### Author Response · Authors · 2025-11-28
> > **Rebuttal Discussions Reminder to Reviewer YmJb**
> >
> > Dear Reviewer YmJb
> >
> > We sincerely appreciate the time and effort you dedicated to reviewing our paper. In response to your concerns, we have conducted additional experiments and provided an in-depth analysis on our method.
> >
> > As the discussion period concludes in six days, we kindly request, if possible, that you review our rebuttal at your convenience. Should there be any further points requiring clarification or improvement, please know that we are fully committed to addressing them promptly. Thank you once again for your invaluable contribution to our research.
> >
> > Warm regards,
> >
> > The Authors

---

### Official Review · Reviewer_Nrqh · 2025-11-01

**Soundness:** 3
**Presentation:** 3
**Contribution:** 1
**Rating:** 2
**Confidence:** 4

**Summary:**

This paper introduces TraceRL, a reinforcement learning framework designed to improve the performance of Diffusion Language Models (DLMs) by aligning their post-training objective with their inference trajectory. The authors argue that existing post-training methods, which often rely on random masking, create a mismatch with the more structured, confidence-based sampling process used during inference. TraceRL is a trajectory-aware framework that uses information from the entire sampling process to optimize the model. The paper also proposes a diffusion-based value model to stabilize training and incorporate process rewards. Using TraceRL, the authors develop a series of models called TraDo, which they show achieve state-of-the-art results on several complex reasoning benchmarks for math and coding, reportedly outperforming strong autoregressive models like Qwen2.5-7B and Llama3.1-8B.

**Strengths:**

- The mismatch between the training objective and the inference process in DLM RL is a well-motivated issue. Aligning the RL training process with the model's actual inference trajectory is logical and well-supported.
- The authors have conducted comprehensive ablation studies that analyze the contribution of different components of their method, such as the value model, the choice of RL optimization strategy, and the shrinkage parameter.

**Weaknesses:**

- Marginal Performance Improvement and Extra Value Model Compared to Autoregressive Models. TraceRL improves previous DLM RL methods, but the performance gain through TraceRL is still marginal (4%\~5% on very simple benchmarks like MATH500). This largely lags behind common autoregressive LM RL recipes like GRPO/DAPO, which could easily bring 10%\~20% gains for a 7B-scale model on MATH500 without any process reward. Post-training is critical to LLMs, while the significance of DLM RL remains questionable given the marginal improvement.
- Incremental Novelty. The paper positions TraceRL as a novel framework that, unlike prior work, uses information "embedded in the sampling trajectory." However, most policy gradient RL methods are inherently "trajectory-aware.", especially the diffusion RL methods like FlowGRPO/DanceGRPO. The objective function in Equation 3 appears to be a standard PPO-style implementation applied to the steps of a diffusion process, which has already been done in FlowGRPO/DanceGRPO. The novelty seems to lie in the specific application to DLMs rather than a fundamental innovation in RL methodology.
- Overclaiming in the Title and Narrative. The title uses the word "Revolutionizing," which sets an exceptionally high bar that the contributions, while solid, do not appear to meet. The work does not fundamentally change the paradigm of diffusion RL.
- Confounding Effect of Long-CoT Fine-Tuning. The TraDo-8B-Thinking model achieves the most impressive results (e.g., 87.4% on MATH500). However, this model is the result of first applying TraceRL and then performing an additional semi-autoregressive fine-tuning step on the OpenThoughts dataset. A significant portion of the performance lift is likely attributable to the long-CoT data, and it is unclear how much TraceRL contributes. Even without TraceRL, the high-quality data of OpenThoughts itself could bring huge performance improvement. Notably, the OpenThinker-v3 7B autoregressive model tuned with the OpenThoughts dataset achieves an AIME2024 score of 69, significantly higher than TraDo-8B-Thinking's 35.5. This further undermines the necessity of using DLM.

**Questions:**

- Could the authors reevaluate the baselines by applying the same in-domain RL fine-tuning to the autoregressive models? What is the advantage of DLM RL compared to autoregressive RL, given that DLM RL is more demanding in implementation and expensive in training?
- What is the inference speed of TraDo-8B-Thinking? Does it use static or dynamic sampling? Does it have any advantages in speed or performance, compared to OpenThinker models, which are also tuned with the OpenThoughts dataset?

---

> ### Author Response · Authors · 2025-11-18
> **Response to Reviewer Nrqh (part 1/2)**
>
> **Q1: Re-evaluate baselines by applying RL to LLM. DLM 4~5% improvement on very simple benchmarks MATH500 is marginal.**
>
> We used prompt batch size 128, 32 responses, and 240 steps (980k total rollouts) on DLM in experiment. Now we run DAPO on Qwen2.5-7B-Instruct using twice the responses (1.96M rollouts). Other parameters remain identical. AR's acc improvement matches that of 8B DLM and is below 4B DLM’s gain. **DLM's substantial improvement with half rollouts highlights the significance of results.**
>
> **Table 1. Acc improvement from RL**
> ||before|after|+Acc%|
> |-|-|-|-|
> |our 4B|70.2|75.6|5.4|
> |our 8B|74.3|78.5|4.2|
> |AR 7B|74.0|78.3|4.3|
>
> **Q2: GRPO/DAPO easily increases 10~20% on MATH500 for 7B LLM.**
>
> This contradicts results in Table 1. Based on our experience, 10–20% gain for 7B LLM occurs only under specific conditions:
> 1. Tremendous computational resources (unaffordable for academic) like DeepSeek-R1-Zero or the DAPO paper. DAPO performs RL on 32B LLM with 40.96M rollouts.
> 2. Currently many RLVR papers report inflated results by improper evaluation: e.g, not using official prompts (causing failed boxed answer extraction) or using base models with instruction prompts, which yields dramatic gains simply due to weakened baseline (much lower than official acc).
>
> **Could reviewer please provide references with open-source code for reproducibility, where DAPO/GRPO improves Qwen2.5-7B-Instruct from the official 74.0 acc to 84~94 on MATH500 under affordable rollouts (1.96M)?**
>
> **Q3: Advantage of DLM RL compared to AR RL?**
>
> **DLM RL is applied to DLMs, while AR RL is applied to LLMs.** AR RL can not be applied on DLM due to different modeling and architecture.
> 1. Based on Table 1, DLM RL uses only half the rollouts of AR RL to achieve comparable or better performance, which demonstrates the significance of our DLM RL results.
> 2. Compared to AR, DLM has advantage in parallel acceleration (Gemini Diffusion, D2F), unified modeling (MMaDA, Lumina), iterative refinement (Edit Flows), and super data learner [2]. RL can enhance model once SFT data is exhausted, making DLM RL vital.
>
> [2]. Diffusion Language Models are Super Data Learners
>
> **Q4: Extra Value Model Compared to LLM.**
>
> 1. We demonstrated utility of value model in stabilizing training through both theoretical analysis and experiments (the reward variance during RL drop from 6.6e-4 to 3.6e-4, about 45.5%). **Therefore, our novel step-wise diffusion value model is not extra.**
> 2. **Moreover, value model is not redundant even for LLMs.** In issue #42 of the 2k+ star open-source reasoning project Open-Reasoner-Zero (ORZ), value model is shown to help stabilize training.
>
> **Q5: Novelty seems to lie in application to DLMs. Trajectory-aware in diffusion RL is proposed in FlowGRPO/DanceGRPO. Novelty is incremental.**
>
> 1. FlowGRPO/DanceGRPO are for visual generation. Continuous diffusions in CV are **fundamentally different** from the discrete diffusions in NLP. Continuous diffusions operate in continuous space of feature embeddings, where denoising follows SDE. Discrete DLMs generate tokens through categorical transitions governed by Markov chain **(also see response 6)**.
> 2. We first demonstrate that reinforcing **confidence-driven sampling traces** of diffusion language model, **rather than purely random traces** induced by the forward continuous diffusion process, is crucial.
> 3. We propose **a novel step-wise, diffusion-based value model** (not appear in FlowGRPO/DanceGRPO) for DLM. We demonstrate its effectiveness in stabilizing training in both experiments and theoretical analysis.
> 4. We introduce the **first chunk-wise generative process reward for DLM**, which can enable user-defined, fine-grained supervision beyond the simple outcome verifiable reward. It also has huge application potential in general RLHF tasks.
> 5. We propose a training acceleration framework with  **novel shrinkage parameter and sliced block training method.** We have validated the effectiveness in Appendix.
> 6. Our novel sliced training and shrinkage techniques accelerate training and establish the **first DLM RL framework applicable to both full/block attention DLM**, demonstrated across different models and tasks.

---

> > ### Author Response · Authors · 2025-11-18
> > **Response to Reviewer Nrqh (part 2/2)**
> >
> > **Q6: Overclaiming in Title and Narrative.**
> >
> > 1. Previous DLM RL directly applied the GRPO objective, omitting trajectory-level information for nearly a year. We first demonstrate with comprehensive experiments that reinforcing **confidence-driven sampling traces** of DLM, **rather than purely random traces** induced by theoretical diffusion forward, is crucial.
> > 2. Secondly, we propose the most comprehensive DLM RL framework, including efficient RLVR, novel diffusion value model and process reward, and generalize to both full/block DLMs, largely facilitating progress in this community.
> >
> > **Q7: Confounding Effect of Long-CoT SFT. OpenThinker AR models stronger.**
> >
> > 1. **This additional experiment shows DLMs can perform long reasoning too.** In all main experiments, RL is directly applied to instruction models with significant gains.
> > 2. We used only 75K samples (1/16 of all) due to limited resources (improves AIME2024 by 23.7%), and Qwen2.5-7B improves 25.4% on same data, demonstrating the potential competitiveness of DLMs compared to AR models for long-CoT reasoning.
> >
> > **Q8: Inference speed of TraDo-8B-Thinking? What sampling method? Advantages compared to OpenThinker AR models?**
> >
> > TraDo-8B-Thinking uses dynamic sampling (**see Table 2 in paper**). For a fair comparison with OpenThinker-v3 7B, we conduct experiment using PyTorch with KV cache (generate.py in supplementary material) on a A100. Table 2 demonstrates superiority of DLM, owing to parallel sampling.
> >
> > **Table 2. Inference time for different number of tokens**
> > |model|length|minutes|
> > |-|-|-|
> > |DLM 8B|1024|0.9|
> > ||2048|1.5|
> > |AR 7B|1024|1.3|
> > ||2048|2.5|
> >
> > We will be appreciated if reviewer can consider increasing the score.

---

> > > ### Author Response · Authors · 2025-11-27
> > > **Rebuttal Discussions Reminder to Reviewer Nrqh**
> > >
> > > Dear Reviewer Nrqh
> > >
> > > We sincerely appreciate the time and effort you dedicated to reviewing our paper. In response to your concerns, we have conducted additional experiments and provided an in-depth analysis on our method.
> > >
> > > As the discussion period concludes in six days, we kindly request, if possible, that you review our rebuttal at your convenience. Should there be any further points requiring clarification or improvement, please know that we are fully committed to addressing them promptly. Thank you once again for your invaluable contribution to our research.
> > >
> > > Warm regards,
> > >
> > > The Authors

---

### Author Response · Authors · 2025-11-18
**General Response**

We sincerely thank all the reviewers for their thorough reviews and valuable feedback. We are glad to hear that the proposed step-wise value model and acceleration framework are considered novel (reviewers YmJb, hx5e), that the trajectory alignment and evaluation on intermediate steps are well-motivated (reviewers YmJb, hx5e, bxFJ), that the results are strong and superior (reviewers hx5e, bxFJ), that the paper is well written and easy to follow (all reviewers), and that the experiments and techniques are comprehensive (reviewers Nrqh, bxFJ, hx5e). **We have revised the manuscript according to the reviewers’ main suggestions, with all changes marked in blue.**

We have addressed all reviewers' concerns through experiments and clarified previous misunderstandings and sources of confusion in this rebuttal. We summarize our main responses to each reviewer below and leave the detailed explanations to the individual responses that follow.

1. We have conducted three ablation studies on the robustness of process reward modeling and clarified the reward assignment approach (reviewer hx5e).
2. We have added quantitative analyses to demonstrate the significance of the diffusion value model (avoiding ambiguity in the curves) and conducted five additional experiments to make the RL results more convincing (reviewer bxFJ).
3. We have clarified the misunderstandings regarding the shrinking approach and the preference-trajectory collection process, and added a comparative discussion with proposed paper (reviewer YmJb).
4. We have conducted DAPO/GRPO on similar scale LLM using the same training set. Our DLM requires only half the number of rollouts to achieve comparable or better results, demonstrating the significance of our method (reviewer Nrqh).

We reply to each reviewer's questions in detail below their reviews. Please kindly check out them. Thank you and please feel free to ask any further questions.

---

### Meta-Review · Area_Chair_AbPQ · 2025-12-29

**Summary:**

This paper successfully aligns diffusion language model training with actual inference trajectories. By achieving SOTA reasoning performance with twice the sample efficiency of autoregressive baselines, it provides an effective contribution to the field. In the rebuttal phase, the authors successfully addressed most concerns. Furthermore, certain review comments were disregarded due to factual inaccuracies or the suspected use of AI-generated content. Therefore, the final decision is to Accept this submission.

**Reviewer Concerns:**

The two reviewers who recommended rejection did not participate in the discussion, and AC believed that most of the concerns had been resolved after reading the authors' response.

**Reviewer Scores:**

AC believes that reviewers Nrqh and YmJb will improve their scores.

---

### Decision · Program_Chairs · 2026-01-26

Accept (Poster)